# Landscape Surrogate: Learning Decision Losses for Mathematical Optimization Under Partial Information

Arman Zharmagambetov[1]    Brandon Amos[1]    Aaron Ferber[2]
Taoan Huang[2]    Bistra Dilkina[2]    Yuandong Tian[1]
[1]FAIR at Meta    [2]University of Southern California
{armanz,bda,yuandong}@meta.com    {aferber,taoanhua,dilkina}@usc.edu

## Abstract

Recent works in learning-integrated optimization have shown promise in settings where the optimization problem is only partially observed or where general-purpose optimizers perform poorly without expert tuning. By learning an optimizer $\mathbf{g}$ to tackle these challenging problems with $f$ as the objective, the optimization process can be substantially accelerated by leveraging past experience. The optimizer can be trained with supervision from known optimal solutions or implicitly by optimizing the compound function $f \circ \mathbf{g}$. The implicit approach may not require optimal solutions as labels and is capable of handling problem uncertainty; however, it is slow to train and deploy due to frequent calls to optimizer $\mathbf{g}$ during both training and testing. The training is further challenged by sparse gradients of $\mathbf{g}$, especially for combinatorial solvers. To address these challenges, we propose using a smooth and learnable *Landscape Surrogate* $\mathcal{M}$ as a replacement for $f \circ \mathbf{g}$. This surrogate, learnable by neural networks, can be computed faster than the solver $\mathbf{g}$, provides dense and smooth gradients during training, can generalize to unseen optimization problems, and is efficiently learned via alternating optimization. We test our approach on both synthetic problems, including shortest path and multidimensional knapsack, and real-world problems such as portfolio optimization, achieving comparable or superior objective values compared to state-of-the-art baselines while reducing the number of calls to $\mathbf{g}$. Notably, our approach outperforms existing methods for computationally expensive high-dimensional problems.

## 1   Introduction

Mathematical optimization problems in various settings have been widely studied, and numerous methods exist to solve them [25, 32]. Although the literature on this topic is immense, real-world applications consider settings that are nontrivial or extremely costly to solve. The issue often stems from uncertainty in the objective or in the problem definition. For example, combinatorial problems involving nonlinear objectives are generally hard to address, even if there are efficient methods that can handle special cases (e.g., $k$-means). One possible approach could be learning so-called *linear surrogate costs* [16] that guide an efficient linear solver towards high quality solutions for the original hard nonlinear problem. This automatically finds a surrogate mixed integer linear program (MILP), for which relatively efficient solvers exist [19]. Another example is the *smart predict+optimize* framework (a.k.a. decision-focused learning) [13, 41] where some problem parameters are unknown at test time and must be inferred from the observed input using a model (e.g., neural nets).

Despite having completely different settings and purposes, what is common among learning surrogate costs, smart predict+optimize, and other integrations of learning and optimization, is the need to learn a certain target mapping to estimate the parameters of a latent optimization problem. This makes the optimization problem well-defined, easy to address, or both. In this work, we draw general

37th Conference on Neural Information Processing Systems (NeurIPS 2023).

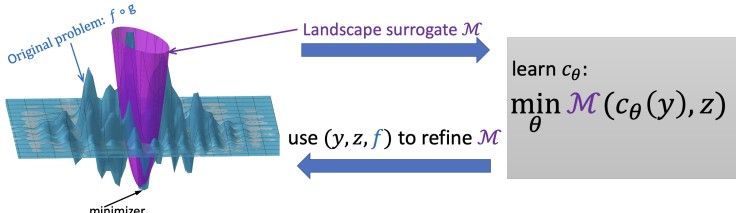

Figure 1: Overview of our proposed framework LANCER. We replace the non-convex and often non-differentiable function $f \circ \mathbf{g}$ with landscape surrogate $\mathcal{M}$ and use it to learn the target mapping $\mathbf{c}_\theta$. The current output of $\mathbf{c}_\theta$ is then used to evaluate $f$ and to refine $\mathcal{M}$. This procedure is repeated in alternating optimization fashion.

connections between different problem families and combine them into a unified framework. The core idea (section 3) is to formulate the learning problem via constructing a compound function $f \circ \mathbf{g}$ that includes a parametric solver $\mathbf{g}$ and the original objective $f$. To the best of our knowledge, this paper is the first to propose a generic optimization formulation (section 3) for these types of problems.

Minimizing this new compound function $f \circ \mathbf{g}$ via gradient descent is a nontrivial task as it requires differentiation through the argmin operator. Although various methods have been proposed to tackle this issue [3, 2], they have several limitations. First, they are not directly applicable to combinatorial optimization problems, which have 0 gradient almost everywhere, and thus require various computationally expensive approximations [34, 41, 15, 39]. Second, even if the decision variables are continuous, the solution space (i.e., argmin) may be discontinuous. Some papers [12, 17] discuss the fully continuous domain but typically involve computing the Jacobian matrix, which leads to scalability issues. Furthermore, in some cases, an explicit expression for the objective may not be given, and we may only have black-box access to the objective function, preventing straightforward end-to-end backpropagation.

These limitations motivate *Landscape Surrogate* losses (LANCER), a unified model for solving coupled learning and optimization problems. LANCER accurately approximates the behavior of the compound function $f \circ \mathbf{g}$, allowing us to use it to learn our target parametric mapping (see fig. 1). Intuitively, LANCER must be differentiable and smooth (e.g., neural nets) to enable exact and efficient gradient computation. Furthermore, we propose an efficient alternating optimization algorithm that jointly trains LANCER and the parameters of the target mapping. Our motivation is that training LANCER in this manner better distills task-specific knowledge, resulting in improved overall performance. Experimental evaluations (section 5) confirm this hypothesis and demonstrate the scalability of our proposed method.

The implementation of LANCER can be found at `https://github.com/facebookresearch/LANCER`.

## 2   Related work

**Smart Predict+Optimize framework** considers settings where we want to train a target mapping which predicts latent components to an optimization problem to improve downstream performance. A straightforward and naive approach in P+O is to build the target machine learning model in a two–stage fashion: train the model on ground truth problem parameters using some standard measure of accuracy (e.g., mean squared error) and then use its prediction to solve the optimization problem. However, this approach is prone to produce highly suboptimal models [6, 41] since the learning problem does not capture the task-specific objectives. Instead, "smart" Predict+Optimize (SPO) [13] proposed to minimize the *decision* regret, i.e., the error induced by the optimization solution based on the estimated problem parameters coming from the machine learning model. A variety of methods exist for learning such models in the continuous, and often convex, optimization setting. These approaches usually involve backpropagating through the solver [11, 12]. In some isolated and simple scenarios, an optimal solution exists [22], or the learning problem can be formulated via efficient surrogate losses [13]. Extending the SPO framework for combinatorial problems is challenging, and current methods rely on identifying heuristic gradients often via continuous, primal, or dual relaxations [33, 41, 34, 31, 39, 15, 28]. Alternative approaches leverage specific structures of the optimization problem [9, 21, 40], such as being solvable via dynamic programming or graph

partitioning. Further work focuses on SPO for individual problems Shah et al. [37], which collects perturbed input instances and learns a separate locally convex loss for each instance. In contrast, we define the loss over the domain rather than per instance, allowing for generalization to unseen instances, and our bilevel optimization formulation enables more efficient training. Moreover, compared to prior works, our method generically applies to a wider variety of problem settings, losses, and problem formulations. See Table 1 for a direct comparison of the requirements and capabilities for different approaches.

**Mixed integer nonlinear programming (MINLP)**  LANCER can be considered as a solver for constrained combinatorial problems with nonlinear and nonconvex objectives (MINLP), a challenging class of optimization problems [4]. Apart from SurCo [16], which we will discuss in detail in section 3, specialized solvers exist that deal with various MINLP formulations [7, 1, 19]. These specialized solvers generally require an analytical form for the objective and cannot handle black-box objectives. Additionally, these methods are not designed to handle general-purpose large-scale problems without making certain approximations or using specialized modeling techniques, such as linearizing the objective function or applying domain-specific heuristics. Lastly, some previous work directly predicts solutions to economic dispatch problems [8], uses reinforcement learning to build solutions to linear combinatorial problems [23], or uses reinforcement learning for ride hailing problems [43]. These approaches are designed for specific application domains or are tailored to linear optimization settings where the reward for a single decision variable is its objective coefficient, which is not trivially applicable in nonlinear settings.

**Optimization as a layer**  The optimization-as-a-layer family of methods considers the composition of functions where one (or more) of the functions is defined as the solution to a mathematical optimization problem [3, 2, 17, 18]. Since both P+O and SurCo can be formulated as a nesting of a target mapping with the `argmin` operator (i.e., optimization solver), one can leverage approaches from this literature. The core idea here is based on using the implicit function theorem to find necessary gradients and backpropagate through the solver. Similar concepts exist for combinatorial optimization [15, 39, 28, 31, 29], where gradient non-existence is tackled through improved primal or dual relaxations.

## 3 A Unified Training Procedure

In this work, we focus on solving the following optimization problems:

$$\min_{\mathbf{x}} f(\mathbf{x}; \mathbf{z}) \qquad \text{s.t.} \quad \mathbf{x} \in \Omega \tag{1}$$

where $f$ is the function to be optimized (linear or nonlinear), $\mathbf{x} \in \Omega$ are the decision variables that must lie in the feasible region, typically specified by (non)linear (in)equalities and possibly integer constraints, and $\mathbf{z} \in \mathcal{Z}$ is the problem description (or problem features). For example, if $f$ is to find a shortest path in a graph, then $\mathbf{x}$ is the path to be optimized, and $\mathbf{z}$ represents the pairwise distances (or features used to estimate them) in the formulation.

Ideally, we would like to have an optimizer that can (1) deal with the complexity of the loss function landscape (e.g., highly nonlinear objective $f$, complicated and possibly combinatorial domain $\Omega$), (2) leverage past experience in solving similar problems, and (3) can deal with a partial information setting, in which only an observable problem description $\mathbf{y}$ can be seen but not the true problem description $\mathbf{z}$ when the decision is made at test time.

To design such an optimizer, we consider the following setting: assume that for the training instances, we have access to the full problem descriptions $\{\mathbf{z}_i\} \subseteq \mathcal{Z}$, as well as the observable descriptions $\{\mathbf{y}_i\} \subseteq \mathcal{Y}$, while for the test instance, we only know its observable description $\mathbf{y}_{\text{test}}$, but not its full description $\mathbf{z}_{\text{test}}$. Note that such a setting naturally incorporates optimization under uncertainty, in which a decision need to be made without full information, while the full information can be obtained in hindsight (e.g., portfolio optimization). Given this setting, we propose the following general *training* procedure on a training set $\mathcal{D}_{\text{train}} := \{(\mathbf{y}_i, \mathbf{z}_i)\}_{i=1}^{N}$ to learn a good optimizer:

$$\min_{\boldsymbol{\theta}} \mathcal{L}(Y, Z) := \sum_{i=1}^{N} f\left(\mathbf{g}_{\boldsymbol{\theta}}(\mathbf{y}_i); \mathbf{z}_i\right) \tag{2}$$

Here $\mathbf{g}_{\boldsymbol{\theta}} : \mathcal{Y} \mapsto \Omega$ is a *learnable solver* that returns a high quality solution for objective $f$ *directly* from the observable problem description $\mathbf{y}_i$. $\boldsymbol{\theta}$ are the learnable solver's parameters. Once $\mathbf{g}_{\boldsymbol{\theta}}$ is learned, we can solve new problem instances with observable description $\mathbf{y}_{\text{test}}$ by either calling $\mathbf{x}_{\text{test}} = \mathbf{g}_{\boldsymbol{\theta}}(\mathbf{y}_{\text{test}})$ to get a reasonable solution, or continue to optimize Eqn. 1 using $\mathbf{x} = \mathbf{x}_{\text{test}}$ as an initial solution.

Theoretically, if problem description $\mathbf{z}$ is fully observable (i.e., $\mathbf{y} = \mathbf{z}$), the optimization oracle $\arg\min_{\mathbf{x}\in\Omega} f(\mathbf{x}; \mathbf{z})$ solves Eqn. 2. However, solving may be computationally intractable even with full information as in nonlinear combinatorial optimization.

Our proposed training procedure is general and covers many previous works that rely on either fully or partially observed problem information.

**Smart Predict+Optimize (P+O)**     In this setting, $f$ belongs to a specific function family (e.g., linear or quadratic programs). The full problem description $\mathbf{z}$ includes objective coefficients, but we only have access to noisy versions of them in $\mathbf{y}$. Then the goal in P+O is to identify a mapping $\mathbf{c}_{\boldsymbol{\theta}}$ (e.g. a neural net) so that a downstream solver outputs a high quality solution: $\mathbf{g}_{\boldsymbol{\theta}}(\mathbf{y}) = \arg\min_{\mathbf{x}\in\Omega} f(\mathbf{x}; \mathbf{c}_{\boldsymbol{\theta}}(\mathbf{y}))$. Here $\arg\min_{\mathbf{x}\in\Omega} f$ can often be solved with standard approaches, and the main challenge is to estimate the problem description accurately (w.r.t. eq. (2)). Note that other P+O formulations can be encompassed within our framework in eq. (2). For instance, the regret-based formulation described in SPO [13] can be represented as $\max_{\hat{\mathbf{z}}\in\mathbf{g}_{\boldsymbol{\theta}}(\mathbf{y})} f(\hat{\mathbf{z}}; \mathbf{z}) - f^*$ where $f^*$ is the optimal loss that is independent of $\boldsymbol{\theta}$.

**Learning surrogate costs for MINLP**     When $f$ is a general nonlinear objective (but $\mathbf{y} = \mathbf{z}$ is fully observed), computing $\arg\min_{\mathbf{x}\in\Omega} f$ also becomes non-trivial, especially if $\mathbf{x}$ is in combinatorial spaces. Such problems are commonly referred as mixed integer nonlinear programming (MINLP). To leverage the power of linear combinatorial solvers, SurCo [16] sets the learnable solver to be $\mathbf{g}_{\boldsymbol{\theta}}(\mathbf{y}) = \arg\min_{\mathbf{x}\in\Omega} \mathbf{x}^{\top}\mathbf{c}_{\boldsymbol{\theta}}(\mathbf{y})$, which is a linear solver and does not include the nonlinear function $f$ at all. Intuitively, this models the complexity of $f$ by the learned *surrogate cost* $\mathbf{c}_{\boldsymbol{\theta}}$, which is parameterized by a neural network. Surprisingly, this works quite well in practice [16].

## 4   LANCER: **Learning** Landscape Surrogate **Losses**

Variations of training objective Eqn. 2 have been proposed to learn $\boldsymbol{\theta}$ in one way or another. This includes derivative-based approaches discussed in section 2 as well as domain-specific methods that learn $\boldsymbol{\theta}$ efficiently and avoid backpropagating through the solver, e.g., SPO+ [13].

While these are valid approaches, at each step of the training process, we need to call a solver to evaluate $\mathbf{g}_{\boldsymbol{\theta}}$, which can be computationally expensive. Furthermore, $\mathbf{g}_{\boldsymbol{\theta}}$ is learned via gradient descent of Eqn. 2, which involves backpropagating through the solver. One issue of this procedure is that the gradient is non–zero only at certain locations (i.e., when changes in the coefficients lead to changes in the optimal solution), which makes the gradient-based optimization difficult.

One question arises: can we model the composite function $f \circ \mathbf{g}_{\boldsymbol{\theta}}$ jointly? The intuition here is that while $\mathbf{g}_{\boldsymbol{\theta}}$ can be hard to compute, $f \circ \mathbf{g}_{\boldsymbol{\theta}}$ can be smooth to model, since $f$ can be smooth around the solution provided by $\mathbf{g}_{\boldsymbol{\theta}}$. If we model $f \circ \mathbf{g}_{\boldsymbol{\theta}}$ locally by a *landscape surrogate* model $\mathcal{M}$, and optimize directly on the local landscape of $\mathcal{M}$, then the target mapping $\mathbf{c}_{\boldsymbol{\theta}}$ can be trained without running expensive solvers:

$$\min_{\boldsymbol{\theta}} \mathcal{M}(Y, Z) := \sum_{i=1}^{N} \mathcal{M}\left(\mathbf{c}_{\boldsymbol{\theta}}(\mathbf{y}_i); \mathbf{z}_i\right). \tag{3}$$

Note that $\mathcal{M}$ directly depends on $\mathbf{c}_{\boldsymbol{\theta}}$ (not on $\mathbf{g}_{\boldsymbol{\theta}}$). Obviously, $\mathcal{M}$ cannot be any arbitrary function. Rather it should satisfy certain conditions: 1) capture a task-specific loss $f \circ \mathbf{g}_{\boldsymbol{\theta}}$ (not just $f$ and not the solver $\mathbf{g}$ alone, but jointly); 2) be differentiable and smooth. Differentiability allows us to train our target model $\mathbf{c}_{\boldsymbol{\theta}}$ in end-to-end fashion (assuming $\mathbf{c}$ is itself differentiable). The primary advantage is that we can *avoid backpropagating through the solver or even through $f$* (e.g., multi-armed bandits). Moreover, $\mathcal{M}_{\mathbf{w}}$ is typically high dimensional (e.g., a neural net) and potentially can make the learning problem for $\mathbf{c}_{\boldsymbol{\theta}}$ much easier. The question is how to obtain such a model $\mathcal{M}$? One

way is to parameterize it and formulate the learning problem:

$$\min_{\mathbf{w}} \sum_{i=1}^{N} \|\mathcal{M}_{\mathbf{w}}(\mathbf{c}_{\boldsymbol{\theta}^*}(\mathbf{y}_i), \mathbf{z}_i) - f\left(\mathbf{g}_{\boldsymbol{\theta}^*}(\mathbf{y}_i); \mathbf{z}_i\right)\|$$

$$\text{s.t.} \quad \boldsymbol{\theta}^* \in \operatorname{argmin}_{\boldsymbol{\theta}} \sum_{i=1}^{N} \mathcal{M}_{\mathbf{w}}(\mathbf{c}_{\boldsymbol{\theta}}(\mathbf{y}_i), \mathbf{z}_i). \tag{4}$$

The main motivation for this bi-level optimization formulation is that the surrogate model $\mathcal{M}_{\mathbf{w}}$ is not concerned to be accurate for all possible $\mathbf{c}_{\boldsymbol{\theta}}$; rather, we are interested in finding the configuration of $\mathcal{M}_{\mathbf{w}}$ that yields the closest approximation to $f \circ \mathbf{g}$ near $\theta^*$. In other words, $\mathcal{M}_{\mathbf{w}}$ serves as a *surrogate loss* that approximates $\mathcal{L}$ at a certain *landscape*: $\mathcal{M}_{\mathbf{w}}(Y, Z; \boldsymbol{\theta}^*) \sim \mathcal{L}(Y, Z; \boldsymbol{\theta}^*)$. By adopting the bi-level optimization framework, we can focus on learning accurate $\mathcal{M}_{\mathbf{w}}$ for such $\theta^*$.

It is important to note that evaluating $f \circ \mathbf{g}$ depends on both $\mathbf{c}_{\boldsymbol{\theta}}(\mathbf{y}_i)$ and $\mathbf{z}_i$ (see eq. (2)). Therefore, to construct an accurate surrogate model $\mathcal{M}_{\mathbf{w}}$ that can effectively approximate $f \circ \mathbf{g}$, it is essential for $\mathcal{M}_{\mathbf{w}}$ to take into consideration the influence of both inputs.

To solve eq. 4, one could apply established methods from the bi-level optimization literature, such as [17, 42]. However, majority of them still rely on $\nabla_{\boldsymbol{\theta}} \mathcal{L}$ (or even $\nabla_{\boldsymbol{\theta}}^2$), which involves differentiating through the solver. To overcome this issue, we propose a simple and generic Algorithm 1, which is based on alternating optimization (high-level idea is depicted in fig. 1). The core idea is to simultaneously learn both mappings ($\mathcal{M}_{\mathbf{w}}$ and $\mathbf{c}_{\boldsymbol{\theta}}$) to explore different solution spaces. By improving our target model $\mathbf{c}_{\boldsymbol{\theta}}$, we obtain better estimates of the surrogate loss around the solution, and a better estimator $\mathcal{M}_{\mathbf{w}}$ leads to better optimization of the desired loss $\mathcal{L}$. The use of alternating optimization helps both mappings reach a common goal. Furthermore, similarly flavored alternating optimization techniques were found to be successful in other problems with non-differentiable nature [44, 45].

In practice, the "inner" optimization (lines 10 and 12) does not have to be done "perfectly". In our approach, we perform a fixed (smaller) number of updates; that is, we do not precisely solve the minimization for both $\mathcal{M}_{\mathbf{w}}$ and $\mathbf{c}_{\boldsymbol{\theta}}$. For example, most experiments use 10-20 updates per $t$. One potential reason is that we do not want the models to overfit to the data; instead, we increase the total number of outer loop $T$. We discuss this further in the experimental setup.

---

**Algorithm 1** Pseudocode for simultaneously learning LANCER and target model $\mathbf{c}_{\boldsymbol{\theta}}$. Note that the algorithm may vary slightly based on setting (e.g., P+O and variations of SurCo ), see Appendix B.

1: Input: $\mathcal{D}_{\text{train}} \leftarrow \{\mathbf{y}_i, \mathbf{z}_i\}_{i=1}^{N}$, solver $\mathbf{g}$, objective $f$, target model $\mathbf{c}_{\boldsymbol{\theta}}$;
2: Initialize $\mathbf{c}_{\boldsymbol{\theta}}$ (e.g. random, warm start);
3: **for** $t = 1 \ldots T$ **do**
4:     • $\mathbf{w}$-step (fix $\boldsymbol{\theta}$ and optimize over $\mathbf{w}$):
5:         **for** $(\mathbf{y}_i, \mathbf{z}_i) \in \mathcal{D}_{\text{train}}$ **do**
6:             evaluate $\hat{\mathbf{c}}_i = \mathbf{c}_{\boldsymbol{\theta}}(\mathbf{y}_i)$;
7:             evaluate $\hat{f}_i = f(\mathbf{g}(\hat{\mathbf{c}}_i); \mathbf{z}_i)$;
8:             add $(\hat{\mathbf{c}}_i, \mathbf{z}_i, \hat{f}_i)$ to $\mathcal{D}$;
9:         **end for**
10:         solve $\min_{\mathbf{w}} \sum_{i \in \mathcal{D}} \left\| \mathcal{M}_{\mathbf{w}}(\hat{\mathbf{c}}_i, \mathbf{z}_i) - \hat{f}_i \right\|$ via supervised learning;
11:     • $\boldsymbol{\theta}$-step (fix $\mathbf{w}$ and optimize over $\boldsymbol{\theta}$):
12:         solve $\min_{\boldsymbol{\theta}} \sum_{i \in \mathcal{D}_{\text{train}}} \mathcal{M}_{\mathbf{w}}(\mathbf{c}_{\boldsymbol{\theta}}(\mathbf{y}_i), \mathbf{z}_i)$ via supervised learning.
13: **end for**

---

Note that the Algorithm 1 avoids backpropagating through the solver or even through $f$. The only requirement is evaluating the function $f$ at the solution of $\mathbf{g}$, which can be achieved by blackbox solver access. As a result, this approach eliminates the complexity and computational expense associated with computing derivatives of combinatorial solvers, making it a more efficient and practical solution as shown in Table 1.

It is worth noting that our framework shares similarities with *actor-critic reinforcement learning* [24]. In this analogy, we can think of $\mathbf{c}_{\boldsymbol{\theta}}$ as an actor responsible for making certain decisions, like predicting coefficients. On the other hand, $\mathcal{M}_{\mathbf{w}}$ plays the role of a critic which evaluates the actor's decisions by estimating the objective value $f$, and provides feedback to the actor.

Table 1: Conceptual comparison of methods from related literature. Capabilities on the left are present or not for Methods on the top. $\pm$ is given when this capability depends empirically on the problem at hand.

| Feature \ Method | LANCER | SurCo [16] | LODLs [37] | SPO+ [13] | DiffOpt [15, 2, 34] | Exact [1, 36] |
|---|---|---|---|---|---|---|
| On-the-fly opt | + | + | − | − | − | + |
| Nonlinear $f$ | + | + | + | − | + | + |
| Blackbox $f$ | + | − | + | − | − | − |
| $\partial f$ not required | + | − | + | + | − | + |
| $\partial \mathbf{g}_{\theta}$ not required | + | − | + | + | − | + |
| Generalization | + | + | − | + | + | − |
| Few fast solver calls | $\pm$ | $\pm$ | − | $\pm$ | $\pm$ | − |

Similar to many other approaches designed for solving problems with bi-level optimizations, including actor-critic algorithms, we currently lack theoretical guarantees regarding the convergence of LANCER with respect to either $\mathcal{M}_{\mathbf{w}}$ or $\mathbf{c}_{\theta}$. This lack of guarantees can also impact the stability of the algorithm. However, this is primarily a matter that can be addressed through empirical investigation and depends on various factors, which is typical in the process of model selection. In Appendix E, we present ablation studies that we conducted to evaluate the stability of our approach across various hyperparameters and neural network architectures. These, along with our main experimental results in section 5, demonstrate that LANCER generally exhibits robustness in many cases.

### 4.1 Reusing landscape surrogate model $\mathcal{M}_{\mathbf{w}}$

Once Algorithm 1 finishes, we usually discard $\mathcal{M}_{\mathbf{w}}$ as it is an intermediate result of the algorithm, and we only retain $\mathbf{c}_{\theta}$ (and solver $\mathbf{g}$) for model deployment. However, we have found through empirical exploration that the learned surrogate loss $\mathcal{M}_{\mathbf{w}}$ can be *reused* for a range of problems, increasing the versatility of the approach. This is particularly advantageous for *SurCo* setting, where we handle one instance at a time. In this scenario, we utilize the *trained* $\mathcal{M}_{\mathbf{w}}$ for unseen test instances by executing only the $\boldsymbol{\theta}$-step of Algorithm 1. The main advantage of this extension is that it eliminates the need for access to the solver $\mathbf{g}$, leading to significant deployment runtime improvements.

### 4.2 Reusing past evaluations of $f \circ \mathbf{g}_{\theta}$

In LANCER, the learning process of $\mathcal{M}_{\mathbf{w}}$ is solely reliant on $\mathcal{D}$ and is independent of the current state of $\mathbf{c}_{\theta}$. Put simply, to effectively learn $\mathcal{M}_{\mathbf{w}}$, we only need the inputs and outputs of $f \circ \mathbf{g}_{\theta}$, namely $\mathbf{c}_{\theta}(\mathbf{y}_i)$, $Z$, and the corresponding objective value $\hat{f}$. Interestingly, we can cache the predicted descriptions themselves, $\mathbf{c}_{\theta}(\mathbf{y}_i)$, without the need for the model $\boldsymbol{\theta}$ or problem information. This caching mechanism allows us to reuse the data $(\mathbf{c}_{\theta}(\mathbf{y}_i), \mathbf{z}, \hat{\mathbf{f}})$ from previous iterations $(1 \ldots T-1)$ as-is. By adopting this practice, we enhance and diversify the available training data for $\mathcal{M}_{\mathbf{w}}$, which proves particularly advantageous for neural networks. This concept bears resemblance to the concept of a *replay buffer* [27] commonly found in the literature on Reinforcement Learning.

## 5 Experiments

We validate our approach (LANCER) in two settings: smart predict+optimize and learning surrogate costs for MINLP. For each setting, we study a range of problems, including linear, nonlinear, combinatorial, and others, encompassing both synthetic and real-world scenarios. Overall, LANCER *exhibits superior or comparable objective values while maintaining efficient runtime*. Additionally, we perform ablation studies, such as re-using $\mathcal{M}$.

### 5.1 Synthetic data

#### 5.1.1 Combinatorial optimization with linear objective

The shortest path (SP) and multidimensional knapsack (MKS) are both classic problems in combinatorial optimization with broad practical applications. In this setting, we consider a scenario where problem parameters $\mathbf{z}$, such as graph edge weights and item prices, cannot be directly observed

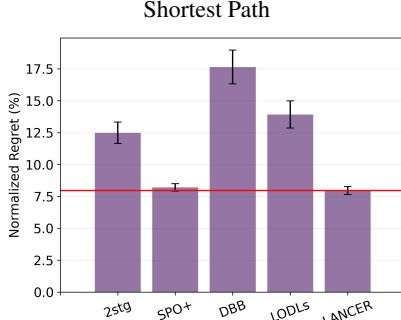
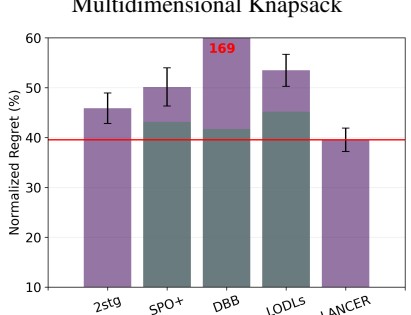

Figure 2: Normalized test regret (lower is better) for different P+O methods: 2-stage, SPO+ [13], DBB [34], LODLs [37] and ours (LANCER). Overlaid dark green bars (right) indicate that the method warm started from the solution of 2stg. DBB performs considerably worse on the right benchmark and is cut off on the $y$-axis.

during test time, and instead need to be estimated from $\mathbf{y}$ via learnable mapping $\mathbf{z} = \mathbf{c}_{\boldsymbol{\theta}}(\mathbf{y})$. That is, we consider smart P+O setting.

**Setup**  We use standard linear program (LP) formulation for SP and mixed integer linear program (MILP) formulation for MKS. Observed features $\mathbf{y}$ and the corresponding ground truth problem descriptions $\mathbf{z}$ for SP are generated using the same procedure as in [38]: grid size of $5 \times 5$, feature dimension of $\mathbf{y} \in \mathbb{R}^5$ (obtained using a random linear mapping from $\mathbf{z}$), 1000 instances for both train and test. For MKS, we increased the knapsack dimension to 5 and capacity to 45, and we set the number of items to 100. Moreover, we use randomly initialized MLP (1 ReLu hidden layer) to generate features of dimension $\mathbf{y} \in \mathbb{R}^{256}$. As for the baselines, apart from the naive 2-stage approach, we have SPO+ [13] and DBB [34], both implemented in PyEPO [38] library. Furthermore, we added LODLs, a novel method from Shah et al. [37]. We explored as best as we could all important hyperparameters for all methods on a fixed cross-validation set. Regarding the LANCER, we utilize MLP with 2 tanh hidden layers of size 200 for surrogate model $\mathcal{M}$. We set $T = 10$ and the number of updates for $\mathcal{M}$ and $\mathbf{c}_{\boldsymbol{\theta}}$ is at most 10. We use SCIP [1] to solve LP for the shortest path and MILP for knapsack. Further details can be found in Appendix C.1.1.

**Results**  The results are summarized in fig. 2. We report the normalized regret as described in [38]. The findings indicate that LANCER and SPO+ consistently outperform the two-stage baseline, particularly when considering the warm start. As SPO+ is specifically designed for linear programs, it provides informative gradients, making it a robust baseline. Even in MKS, where theorems proposed in [13] are no longer applicable, SPO+ performs decently with minimal tuning effort. The DBB approach, however, demonstrates unsatisfactory default performance but can yield favorable outcomes with proper initialization and tuning (see the right plot). Interestingly, the other P+O baselines, initialized randomly, were unable to outperform a naive 2stg in both benchmarks.

LANCER achieves superior performance in both tasks, with a noticeable advantage in MKS. This may be attributed to the high dimension of the MKS problem and the large feature space ($\mathbf{y}$). One possible explanation is that the sparse gradients of the derivative-based method make the learning problem harder, whereas LANCER models the landscape of $f \circ g$, providing informative gradients for $\mathbf{c}_{\boldsymbol{\theta}}$.

### 5.1.2 Combinatorial optimization with nonlinear objective

In this section, we apply LANCER for solving mixed integer nonlinear programs (MINLP). Specifically, we transform a combinatorial problem with a nonlinear objective into an instance of MILP via learning linear surrogate costs as described in Ferber et al. [16]. Note that in this setting, we assume that the full problem description $\mathbf{y} = \mathbf{z}$ is given and fully observable (in contrast to the P+O setting).

We begin by examining on-the-fly optimization, where each problem is treated independently. In this scenario, the cost vector $\mathbf{c}_{\boldsymbol{\theta}}(\mathbf{y})$ simplifies to a constant value $\mathbf{c}$. SurCo is then responsible for directly training the cost vector $\mathbf{c}$ of the linear surrogate. As we lack a distribution of problems to train $\mathcal{M}$, specific adaptations to Algorithm 1 are necessary, which are outlined in detail in Appendix B. We refer to this version as LANCER–zero to be consistent with SurCo–zero.

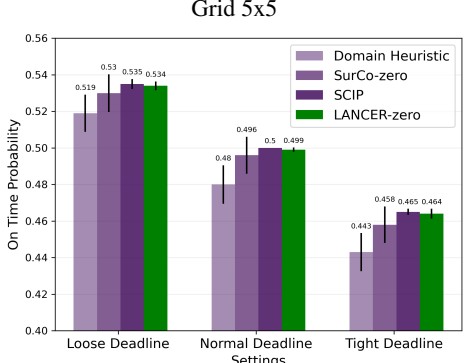
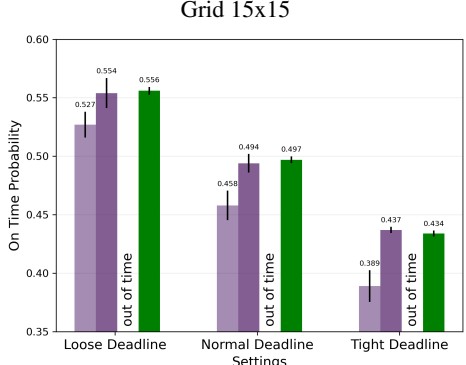

Figure 3: Results on stochastic shortest path using different grid sizes: 5x5 (left) and 15x15 (right). We report avg objective values (higher is better) on three settings described in [16]. For grid size of 15x15, SCIP [1] was unable to finish within the 30 min time limit.

**Setup** Nonlinear shortest path problems arise when the objective is to maximize the probability of reaching a destination before a specified time in graphs with random edges [26, 14]. The problem formulation is similar to the standard linear programming (LP) formulation of the shortest path, as described in section 5.1.1, with a few adjustments: 1) the weight of each edge follows a normal distribution, i.e., $w_e \sim \mathcal{N}(\mu_e, \sigma_e)$; 2) the objective is to maximize the probability that the sum of weights along the shortest path is below a threshold $W$, which can be expressed using the standard Gaussian cumulative distribution function (CDF), $P(\sum_{e \in E} w_e \leq W) = \Phi\left((W - \sum_{e \in E} \mu_e)/\sqrt{\sum_{e \in E} \sigma_e}\right)$ where $E$ is the set of edges belonging to the shortest path. We use $5 \times 5$ and $15 \times 15$ grid graphs with 25 draws of edge weights. We set the threshold $W$ to three different values corresponding to loose, normal, and tight deadlines. The remaining settings are adapted from Ferber et al. [16], and additional details can be found in Appendix C.1.2.

**Results** Fig. 3 illustrates the performance of different methods in both grid sizes. SCIP directly formulates the MINLP to maximize the CDF, resulting in an optimal solution. However, this approach is not scalable for larger problems and is limited to smaller instances like the $5 \times 5$ grid. The heuristic method assigns each edge weight as $w_e = \mu_e + \gamma\sigma_e$, where $\gamma$ is a user-defined hyperparameter, and employs standard shortest path algorithms (e.g., Bellman-Ford). As the results indicate, this heuristic approach produces highly suboptimal solutions. SurCo–zero and LANCER–zero demonstrate similar performance, with LANCER–zero being superior in almost all scenarios.

## 5.2 Real-world use case: quadratic and broader nonlinear portfolio selection

### 5.2.1 The quadratic programming (QP) formulation

In this study, we tackle the classical quadratic Markowitz [30] portfolio selection problem. We use real-world data from Quandl [35] and follow the setup described in Shah et al. [37]. The prediction task leverages each stock's historical data **y** to forecast future prices **z**, which are then utilized to solve the QP (i.e., P+O setting). The predictor is the MLP with 1 hidden layer of size 500.

Table 2: Portfolio selection normalized test decision loss (lower is better).

| Method | Test DL |
|---|---|
| Random | 1 |
| Optimal | 0 |
| 2–Stage | $0.57 \pm 0.02$ |
| LODLs [37] | $0.55 \pm 0.02$ |
| MDFL [41] | $0.52 \pm 0.01$ |
| **LANCER** | $\mathbf{0.53 \pm 0.02}$ |

**Setup** We follow a similar setup described in [37], except for a fix in the objective's quadratic term that slightly affects the decision error's magnitude. More details can be found in Appendix C.2.1. We compare LANCER against two-stage and LODLs. However, SPO+ is not applicable in this nonlinear setting, and DBB's performance is notably worse since it is designed for purely combinatorial problems. Additionally, we report the optimal solution (using the ground truth values of **z**) and the (Melding Decision Focused Learning) MDFL [41] method, which leverages the implicit function theorem to differentiate through the KKT conditions. For our LANCER implementation, we use an MLP with 2 hidden layers for $\mathcal{M}$

and update each of the parametric mappings 5 times per iteration with a total of $T = 8$ iterations. We use cvxpy [10] to solve the QP.

**Results**   Table 2 summarizes our results. We report the normalized decision loss (i.e., normalized Eqn. (2)) on test data. Since the problem is smooth and exact gradients can be calculated, MDFL achieves the best performance closely followed by the LANCER. The remaining results are in agreement with [37]. While LANCER does not achieve the best overall performance, it does so using a significantly smaller number of calls to a solver, as we discuss in more detail in Section 5.3.

### 5.2.2   Combinatorial portfolio selection with third-order objective

The convex portfolio optimization problem discussed in the previous section 5.2.1 is unable to capture desirable properties such as logical constraints [5, 15], or higher-order loss functions [20] that integrate metrics like co-skewness to better model risk. We use the Quandl [35] data (see Appendix C.2.2 for details on setup) and similar to section 5.1.2, we assume that the full problem description **z** is given at train/test time.

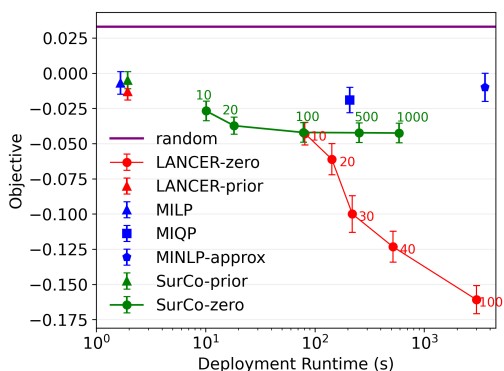

**Results** are shown in fig. 4. We first tried to solve the given MINLP exactly via SCIP. However, it fails to produce the optimal solution within a 1 hour time limit and we report the best incumbent feasible solution. MIQP (blue squares) and MILP (blue triangles) approximations overlook the co-skewness and non-linear terms, respectively. Comparing their performance, MIQP exhibits a $2\times$ lower loss than the MILP baseline but in a significantly longer

Figure 4: Objective (lower is better) and deployment runtime for combinatorial portfolio selection problem. For LANCER–zero and SurCo–zero, numbers at each point correspond to the number of iterations.

runtime. For LANCER and SurCo, we present results for two scenarios: learning the linear cost vector **c** directly (*zero*) for each instance, and a parameterized version $\mathbf{c}_\theta(\mathbf{z})$ (*prior*). The main distinction of "prior" is that no learning occurs during test time, as we directly map the problem descriptor **z** to a linear cost vector and solve the MILP. Consequently, the deployment runtime is similar to that of the MILP approximation, but LANCER–prior produces slightly superior solutions. Remarkably, LANCER–zero achieves significantly better loss values, surpassing all other methods. Although it takes longer to run, the runtime remains manageable, and importantly, the solution quality improves with an increasing number of iterations.

According to our experimental findings, LANCER demonstrated the most substantial improvement in this specific setting. Encouraged by this success, we conducted a deeper exploration of the complexities of this problem, as detailed in Appendix D.

### 5.3   Computational efficiency

Comparing baseline methods, including LANCER, we find that querying solver $\mathbf{g}_\theta$ is the primary computational bottleneck. To evaluate this aspect, we empirically analyze different algorithms on various benchmarks in the P+O domain. The results, depicted in fig. 5, highlight that LODLs require sampling a relatively large number of points per training instance, leading to potentially time-consuming solver access. On the other hand, gradient-based methods like DBB, MDFL, and SPO+ typically solve the optimization problem 1-2 times per update but require more iterations to converge. In contrast, LANCER accesses the solver in the **w**-step, with the number of accesses proportional to the training set size and a small total number of alternating optimization iterations. Moreover, we leverage saved solutions from previous iterations, akin to a replay buffer, when fitting $\mathcal{M}$. These combined factors allow us to achieve favorable results with a small value of $T$.

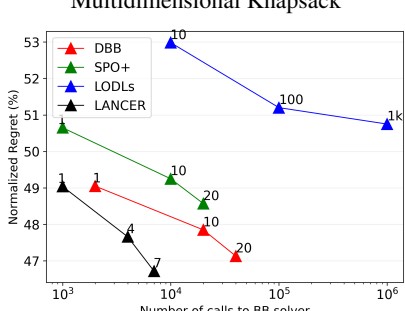
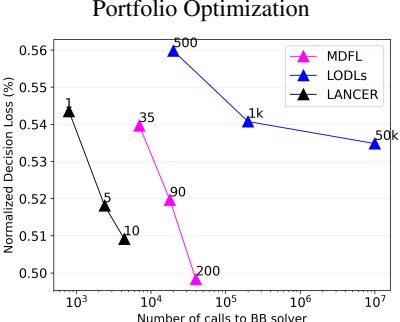

Figure 5: Trade-off curves between black-box solver calls (MILP or QP) vs decision loss (or regret) on P+O problems. Point labels (e.g. 1,4,7) correspond to the epoch; except for LODLs, where they correspond to the number of samples per instance. Each algorithm uses a different number of BB calls per epoch.

Table 3: Results of reusing $\mathcal{M}$ on stochastic shortest path problem from fig. 3 ($15 \times 15$ grid). Here, "reused $\mathcal{M_w}$" has limited access to the solver $\mathbf{g}$, and thus is much faster while retaining solution quality.

| Method | Loose Deadline | | Normal Deadline | | Tight Deadline | |
|---|---|---|---|---|---|---|
| | obj. | time (s) | obj. | time (s) | obj. | time (s) |
| LANCER–zero | $0.556 \pm 0.006$ | $61.1 \pm 3.2$ | $0.497 \pm 0.004$ | $62.3 \pm 2.9$ | $0.434 \pm 0.005$ | $62.8 \pm 2.7$ |
| LANCER–reused $\mathcal{M_w}$ | $0.556 \pm 0.007$ | $2.9 \pm 0.3$ | $0.496 \pm 0.004$ | $2.7 \pm 0.6$ | $0.432 \pm 0.004$ | $2.5 \pm 0.6$ |

## 5.4 Reusing landscape surrogate $\mathcal{M_w}$

In this scenario, we introduce a dependency of $\mathcal{M_w}$ on both the predicted linear cost ($\mathbf{c}$) and the problem descriptor ($\mathbf{y}$), as described in section 4.1. This enables us to reuse $\mathcal{M_w}$ for different problem instances without retraining, and eliminate the dependency on the solver $\mathbf{g}_\theta$, giving LANCER a substantial runtime acceleration. To validate this hypothesis, we pretrain $\mathcal{M_w}$ using 200 instances of the stochastic shortest path on a $15 \times 15$ grid by providing concatenated $(\mathbf{c}, \mathbf{y})$ as input. We apply LANCER–zero to the same test set as before and present results in fig. 3, demonstrating comparable performance between these two approaches, with "reused $\mathcal{M_w}$" being much faster.

## 6 Conclusion

This paper makes a dual contribution. Firstly, we derive a unified training procedure to address various coupled learning and optimization settings, including smart predict+optimize and surrogate learning. This is significant as it advances our understanding of learning-integrated optimization under partial information. Secondly, we propose an effective and powerful method called LANCER to tackle this training procedure. LANCER offers several advantages over existing literature, such as versatility, differentiability, and efficiency. Experimental results validate these advantages, leading to significant performance improvements, especially in high-dimensional spaces, both in problem description and feature space. One potential drawback is the complexity of tuning $\mathcal{M}$, requiring model selection and training. However, future research directions include addressing this drawback and exploring extensions of LANCER, such as applying it to fully black box $f$ scenarios.

## Acknowledgments and Disclosure of Funding

The research at the University of Southern California was supported by the National Science Foundation (NSF) under grant number 2112533. We also thank the anonymous reviewers for helpful feedback.

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

# A  Computational complexity

It is quite straightforward to estimate the runtime of our approach from Algorithm 1 in the main paper. Let us denote $I_{\mathbf{g}}$ as the time to get the solution from solver $\mathbf{g}(\mathbf{y}_i)$. Also, let $I_{\mathbf{w}}$ and $I_{\boldsymbol{\theta}}$ be the runtime of training the parametric loss $\mathcal{M}_{\mathbf{w}}$ and the target model $\mathbf{c}_{\boldsymbol{\theta}}$, respectively. Then, assuming evaluating $f$ is negligible, one iteration of our algorithm naively takes $\mathcal{O}(N \cdot I_{\mathbf{g}} + I_{\mathbf{w}} + I_{\boldsymbol{\theta}})$, so the total runtime is $\mathcal{O}(T \cdot N \cdot I_{\mathbf{g}} + T \cdot I_{\mathbf{w}} + T \cdot I_{\boldsymbol{\theta}})$. In practice, we iterate at most 100 times ($T < 100$). Therefore, the main bottleneck is $\mathcal{O}(T \cdot N \cdot I_{\mathbf{g}})$, which is mainly overtaken by an access to $\mathbf{g}$. Although we claim that leveraging $\mathcal{M}$ to learn $\boldsymbol{\theta}$ is efficient, we admit that there is still a requirement to access the solver $\mathbf{g}$.

However, there are several accelerations that can be made:

1. $\mathcal{O}(N \cdot I_{\mathbf{g}})$ is embarrassingly parallel computation since each request to $\mathbf{g}$ is independent and, additionally, subsampling on $\mathcal{D}_{\text{train}}$ can be performed.

2. We can "warm start" the solver $\mathbf{g}$ from solutions obtained in $t-1$, which typically yields faster convergence.

3. Although we did not test this, but one can "early stop" the solver $\mathbf{g}$ if it is too costly to solve optimally (e.g. large scale MILP). Our hypothesis is that it is enough to obtain a feasible solution $\hat{\mathbf{x}}$ in certain neighborhood of $\mathbf{x}^*$. Since we are still able to evaluate $f$ and $(f(\hat{\mathbf{x}}), \hat{\mathbf{c}}, \mathbf{z})$ is a "valid" tuple, we can use it to train $\mathcal{M}_{\mathbf{w}}$.

4. Moreover, we empirically found out that the supervised learning steps (lines 10 and 12) do not require "perfect" learning. That is, we perform several gradient updates over $\mathbf{w}$ and $\boldsymbol{\theta}$, which significantly reduces $I_{\mathbf{w}}$ and $I_{\boldsymbol{\theta}}$.

# B  Pseudocodes

Algorithms 2-4 below closely resemble the Algorithm 1 in the main paper. However, there are minor variations that depend on the problem setting, whether it involves learning linear surrogates for MINLP or smart Predict+Optimize setting.

---

**Algorithm 2** Pseudocode for learning LANCER and target model $\mathbf{c}_{\boldsymbol{\theta}}$ for *smart Predict+Optimize* setting. Note: $\mathbf{y}$ – input (always observed) features, $\mathbf{z}$ – ground truth problem descriptions (available at train time only).

---

1: Input: $\mathcal{D}_{\text{train}} \leftarrow \{\mathbf{y}_i, \mathbf{z}_i\}_{i=1}^{N}$, solver $\mathbf{g}$, objective $f$ (can be black–box), target model $\mathbf{c}_{\boldsymbol{\theta}}$, (optional) prediction loss penalty $\lambda$;
2: Initialize $\mathbf{c}_{\boldsymbol{\theta}}$ from solving: $\min_{\boldsymbol{\theta}} \sum_{i \in \mathcal{D}_{\text{train}}} \|\mathbf{c}_{\boldsymbol{\theta}}(\mathbf{y}_i) - \mathbf{z}_i\|$;
3: Set $\mathcal{D} \leftarrow \{\}$;
4: **for** $t = 1 \ldots T$ **do**
5:    • $\mathbf{w}$-step (fix $\boldsymbol{\theta}$ and optimize over $\mathbf{w}$):
6:       **for** $(\mathbf{y}_i, \mathbf{z}_i) \in \mathcal{D}_{\text{train}}$ **do**
7:          evaluate $\hat{\mathbf{c}}_i = \mathbf{c}_{\boldsymbol{\theta}}(\mathbf{y}_i)$;
8:          evaluate $\hat{f}_i = f(\mathbf{g}(\hat{\mathbf{c}}_i); \mathbf{z}_i)$;
9:          add $(\hat{\mathbf{c}}_i, \mathbf{z}_i, \hat{f}_i)$ to $\mathcal{D}$;
10:       **end for**
11:       minimize: $\min_{\mathbf{w}} \sum_{j \in \mathcal{D}} \left\| \mathcal{M}_{\mathbf{w}}(\hat{\mathbf{c}}_j, \mathbf{z}_j) - \hat{f}_j \right\|$;
12:    • $\boldsymbol{\theta}$-step (fix $\mathbf{w}$ and optimize over $\boldsymbol{\theta}$):
13:       minimize: $\min_{\boldsymbol{\theta}} \sum_{i \in \mathcal{D}_{\text{train}}} \mathcal{M}_{\mathbf{w}}(\mathbf{c}_{\boldsymbol{\theta}}(\mathbf{y}_i), \mathbf{z}_i) + \lambda \|\mathbf{c}_{\boldsymbol{\theta}}(\mathbf{y}_i) - \mathbf{z}_i\|$.
14: **end for**

---

**Algorithm 3** Pseudocode for learning linear surrogates with LANCER–zero (used in sections 5.1.2 and 5.2.2). Note that $\mathbf{y} = \mathbf{z}$ in this setting and we have only one optimization problem instance.

---

1: **Input:** problem description $\mathbf{y}$, solver $\mathbf{g}$, objective $f$ (can be black–box);
2: Initialize $\mathbf{c} \in \mathbb{R}^L$;
3: Set $\mathcal{D} \leftarrow \{\}$;
4: **for** $t = 1 \ldots T$ **do**
5:     • $\mathbf{w}$-step (fix $\mathbf{c}$ and optimize over $\mathbf{w}$):
6:        randomly sample $\{\hat{\mathbf{c}}_i\}_i^N$ around $\mathbf{c}$;
7:        **for** $i = 1 \ldots N$ **do**
8:           evaluate $\hat{f}_i = f(\mathbf{g}(\hat{\mathbf{c}}_i); \mathbf{y})$;
9:           add $(\hat{\mathbf{c}}_i, \hat{f}_i)$ to $\mathcal{D}$;
10:        **end for**
11:        minimize: $\min_{\mathbf{w}} \sum_{j \in \mathcal{D}} \left\| \mathcal{M}_{\mathbf{w}}(\hat{\mathbf{c}}_j) - \hat{f}_j \right\|$;
12:     • $\boldsymbol{\theta}$-step (fix $\mathbf{w}$ and optimize over $\mathbf{c}$):
13:        // $\boldsymbol{\theta} = \mathbf{c}$ in this setting as we solve for a single problem instance $\mathbf{y}$
14:        minimize: $\min_{\mathbf{c}} \mathcal{M}_{\mathbf{w}}(\mathbf{c})$.
15: **end for**

---

**Algorithm 4** Pseudocode for learning linear surrogates with LANCER–prior (used in section 5.2.2): we learn $\mathbf{c}_{\boldsymbol{\theta}}$ on a distribution of optimization problems. Note that $\mathbf{y} = \mathbf{z}$ in this setting

---

1: **Input:** $\mathcal{D}_{\text{train}} \leftarrow \{\mathbf{y}_i\}_{i=1}^N$, solver $\mathbf{g}$, objective $f$ (can be black–box), target model $\mathbf{c}_{\boldsymbol{\theta}}$;
2: Initialize $\mathbf{c}_{\boldsymbol{\theta}}$ (random, warm start from heuristics);
3: Set $\mathcal{D} \leftarrow \{\}$;
4: **for** $t = 1 \ldots T$ **do**
5:     • $\mathbf{w}$-step (fix $\boldsymbol{\theta}$ and optimize over $\mathbf{w}$):
6:        **for** $(\mathbf{y}_i) \in \mathcal{D}_{\text{train}}$ **do**
7:           evaluate $\hat{\mathbf{c}}_i = \mathbf{c}_{\boldsymbol{\theta}}(\mathbf{y}_i)$;
8:           evaluate $\hat{f}_i = f(\mathbf{g}(\hat{\mathbf{c}}_i); \mathbf{y}_i)$;
9:           add $(\hat{\mathbf{c}}_i, \mathbf{y}_i, \hat{f}_i)$ to $\mathcal{D}$;
10:        **end for**
11:        minimize: $\min_{\mathbf{w}} \sum_{j \in \mathcal{D}} \left\| \mathcal{M}_{\mathbf{w}}(\hat{\mathbf{c}}_j, \mathbf{y}_j) - \hat{f}_j \right\|$;
12:     • $\boldsymbol{\theta}$-step (fix $\mathbf{w}$ and optimize over $\boldsymbol{\theta}$):
13:        minimize: $\min_{\boldsymbol{\theta}} \sum_{i \in \mathcal{D}_{\text{train}}} \mathcal{M}_{\mathbf{w}}(\mathbf{c}_{\boldsymbol{\theta}}(\mathbf{y}_i), \mathbf{y}_i)$.
14: **end for**

---

## C  Details of experimental setup

### C.1  Synthetic data

#### C.1.1  Combinatorial optimization with linear objective

**Data**  We follow the setup and scripts from PyEPO [38] library to generate data.

- For SP, we follow the same data generation process as in the original scripts: $5 \times 5$ grid (40 total edges), 1000 training problem instances with 5 input features (i.e., $Y \in \mathbb{R}^{1000 \times 5}$) and the same for the test set. Input features ($\mathbf{y}$) are generated using normal distribution with $\mathcal{N}(\mathbf{0}, \mathbf{1})$. The ground truth descriptors $\mathbf{z}$ are obtained by first randomly and linearly projecting $\mathbf{y}$ in 40 dimensions followed by nonlinearity (polynomial of degree 6 and normalization). Lastly, random noise is added to $\mathbf{z}$ to make the problem harder. We use the standard linear program (LP) formulation of the shortest path and implement solver $\mathbf{g}$ in SCIP [1].

- As for MKS, we begin by generating a cost vector for each item using a random uniform distribution between 0 and 5. Then, we obtain features $\mathbf{y}$ by passing $\mathbf{z}$ through a random neural network with one hidden layer of size 500 and tanh activation. Knapsack capacity is 40, knapsack dimension is 5, 100 total items to choose from, feature dimension is 256 and there are 1000 instances in both train/test. Lastly, weight for each item is generated

according to the uniform distribution between 0 and 1. We use the standard mixed-integer linear program (MILP) formulation of the multidimensional knapsack and implement solver **g** in SCIP [1].

**Target mapping $c_\theta$**   All baselines use the same target mappings for each problem: linear mapping for SP and MLP with 1 hidden layer for MKS (size of 300 and tanh activation).

**LANCER**   The training procedure closely follows Algorithm 2. Other problem dependent settings are as follows:

- For SP, LANCER uses MLP with 2 hidden layers of size 100 (tanh activation). We train LANCER for $T = 10$ iterations. At each iteration, we make 5 updates (using Adam optimizer with lr= 0.001) for each mappings ($c_\theta$ and $\mathcal{M}_\mathbf{w}$).
- For MKS, LANCER uses MLP with 2 hidden layers of size 200 (tanh activation). We train LANCER for $T = 7$ iterations. At each iteration, we perform 5 updates (using Adam optimizer with lr= 0.001) for $c_\theta$ and 10 updates for $\mathcal{M}_\mathbf{w}$.

**Baselines**

- **SPO+,DBB:** We use the versions implemented within PyEPO and set the number of epochs to 25. Both methods use the Adam optimizer with learning rates tuned for each problem. Other arguments follow the default setting suggested by authors.
- **LODLs:** We use the implementation provided by authors in [37]. We set the number of sampling points to 1000 and employ "random Hessian" version of the algorithm. Other arguments follow the default setting suggested by authors.

### C.1.2   Combinatorial optimization with nonlinear objective

We aim to solve the following optimization problem with nonlinear objective:

$$
\begin{aligned}
\min_{\mathbf{x}} \ & \Phi\left( \left(W - \sum_{(u,v)\in E} x_{u,v}\mu_{u,v}\right) \Big/ \sqrt{\sum_{(u,v)\in E} x_{u,v}\sigma_{u,v}} \right) \\
\text{s.t.} \ & \sum_{(u,t)\in E} x_{u,t} \geq 1 \quad \text{(at least one unit of flow into } t) \\
& \sum_{u:(u,v)\in E} x_{u,t} - \sum_{u:(v,u)\in E} x_{u,t} \geq 0 \quad , \quad \forall v \notin \{s,t\} \quad \text{(flow in} \geq \text{flow out)} \\
& \mathbf{x} \geq 0
\end{aligned}
\tag{5}
$$

where $\Phi$ is the standard Gaussian cumulative distribution function (CDF), $E$ is the set of all edges, $W$ is the user-defined threshold, $\mu_{u,v}$ and $\sigma_{u,v}$ are the mean and the variance of the corresponding edge's weight.

**Data**   We closely follow the data generation process described in [16]. Specifically, each edge is a random variable with $\mu$ coming from the uniform distribution (between 0.1 and 0.2); and with $\sigma$ is also generated uniformly randomly (between 0.1 and 0.3 multiplied by $1 - \mu$). For *thresholds $W$*, we set them as follows: calculate the distance of the shortest path distance using the mean value as an edge weight and multiply it to 0.9 for tight, 1.0 for normal and 1.1 for loose deadlines, respectively. We set the number of problem instances to 25 and use the Bellman-Ford algorithm for a solver **g**.

**LANCER**   The training procedure closely follows Algorithm 3. We use MLP with 2 hidden layers of size 200 (300 for $15 \times 15$ grid size) and tanh activation. We train LANCER for $T = 40$ iterations. At each iteration, we make 10 updates (using Adam optimizer) for each learnable models (**c** and $\mathcal{M}_\mathbf{w}$).

**Baselines**

- **SCIP** attempts to solve the original MINLP defined in Eqn. (5). We set the maximum time limit to 30 min before terminating the solver.

- **Domain Heuristic** simply assigns each edge's weight as $w_e = \mu_e + \gamma \sigma_e$ (where $\gamma$ is a hyperparameter) and run the Bellman-Ford's algorithm.
- **SurCo** [16] replicates the same experimental setup as described in the original paper. Specifically, we use DBB [34] to differentiate through the solver with $\lambda = 1000$ and apply Adam optimizer for 40 epochs with the learning rate $= 0.1$.

## C.2   Real-world use case: quadratic and broader nonlinear portfolio selection

### C.2.1   The quadratic programming (QP) formulation

We use the standard quadratic program formulation of the Markowitz' [30] portfolio selection problem:

$$
\min_{\mathbf{x}} \alpha \mathbf{x}^T \mathbf{G} \mathbf{x} - \boldsymbol{\mu}^T \mathbf{x}
$$
$$
\text{s.t.} \quad \sum_{i=1}^{k} x_i = 1 \quad \text{and} \quad \mathbf{x} \geq 0 \tag{6}
$$

where $\boldsymbol{\mu}$ is an expected return vector for each portfolio and $\mathbf{G}$ is the covariance matrix. We set the user-defined hyperparameter $\alpha = 0.1$ in all experiments and use CvxPy [10] to solve QP in eq. (6).

**Data**   We reuse the code from Shah et al. [37] to generate data (downloaded from QuandlWIKI [35]) and use the same setup. The features $\mathbf{y}$ are historical stock prices and the task is to predict an expected return $\boldsymbol{\mu}$ (using MLP with 1 hidden layer of size 500) in smart Predict+Optimize fashion. There are 200 instances in train/validation set and 400 instances in test set. The number of portfolios in each instance is 50.

**LANCER**   The training procedure closely follows Algorithm 2. We use MLP with 2 hidden layers of size 100 and tanh activation. We train `LANCER` for $T = 8$ iterations. At each iteration, we make 10 updates (using Adam optimizer) to fit each mapping ($\mathbf{c}_{\boldsymbol{\theta}}$ and $\mathcal{M}_{\mathbf{w}}$).

**Baselines**

- **LODLs:** We use the implementation provided by authors in [37]. We replicate their configurations for this experiment, except we made a quick fix in the code as they forgot to add a matrix transpose in the quadratic term.
- **MDFL** (Melding Decision Focused Learning) [41]: we use the version implemented in [37] and follow their configurations.

### C.2.2   Combinatorial portfolio selection with third-order objective

We extend the portfolio selection problem in eq. (6) as follows:

$$
\min_{\mathbf{x}, \mathbf{v}} \alpha \mathbf{x}^T \mathbf{G} \mathbf{x} + \gamma \|\mathbf{x} - \mathbf{x}_0\|_1 - \boldsymbol{\mu}^T \mathbf{x} - \beta \mathbf{x}^T \mathbf{S} \mathbf{x} \otimes \mathbf{x}
$$
$$
\text{s.t.} \quad \sum_{i=1}^{k} x_i = 1
$$
$$
f_{\min} * v_i \leq x_i \leq f_{\max} * v_i \quad \text{for} \quad i = 1 \ldots k \quad \text{(fraction of each selected portfolio)} \tag{7}
$$
$$
m \leq \sum_{i=1}^{k} v_i \leq M \quad \text{(no. of selected portfolios must be between } m \text{ and } M\text{)}
$$
$$
\mathbf{x} \geq 0
$$
$$
\mathbf{v} \in \{0, 1\}
$$

where $\mathbf{S}$ is co-skewness matrix and $\mathbf{v}$ is the binary variables to enforce discrete constraints (e.g. hard limit on number of portfolios). We also introduce the initial portfolio selection vector $\mathbf{x}_0$ (generated uniformly at random) and enforce our final solution to be close to it (via $\gamma$). We set the penalty on co-skewness as $\beta = 0.5$, $\gamma = 0.01$, $m = 3$, $M = 10$, $f_{\min} = 0.01$ and $f_{\max} = 0.2$ throughout all

experiments. We reuse the same data as in section C.2.1 but increase the number of portfolios to 100 and add co-skewness matrix $\mathbf{S}$ to it. Note that in this task we assume that all problem descriptors are fully observed (e.g. $\boldsymbol{\mu}, \mathbf{G}, \mathbf{S}$).

**LANCER** The training procedure for LANCER-zero closely follows Algorithm 3 and LANCER-prior closely follows Algorithm 4. We use MLP with 2 hidden layers of size 300 and tanh activation. We train LANCER for $T = 40$ iterations. At each iteration, we perform 10 updates (using Adam optimizer) for each learnable models ($\mathbf{c}_{\boldsymbol{\theta}}$ and $\mathcal{M}\mathbf{w}$). Additionally, LANCER-prior uses a parametric mapping for $\mathbf{c}_{\boldsymbol{\theta}}$ which is implemented via 1 hidden layer MLP of size 500, which $\boldsymbol{\mu}$ takes as input.

**Baselines**

- **SCIP** attempts to solve the problem defined in Eqn. (7) in three different settings related to the objective: ignores all nonlinear terms (MILP), ignores cubic term (MIQP) and attempts to solve the original nonlinear problem (MINLP). We set the maximum time limit to 60 min before terminating the solver and obtaining the best solution.
- For **SurCo** [16], we use two versions: SurCo-prior and SurCo-zero. SurCo-prior uses the same mapping for $\mathbf{c}_{\boldsymbol{\theta}}$ as LANCER described above. We use DBB [34] to differentiate through the solver with $\lambda = 100$ and apply Adam optimizer for 100 epochs with the learning rate $= 0.1$ (0.0001 for prior).

# D  Combinatorial portfolio selection with third-order objective

Based on our experimental findings, LANCER showcased the most significant improvement in addressing the combinatorial portfolio selection problem. Inspired by this success, we delved further into the intricacies of this problem and sought to visualize the actual solutions obtained by the two best-performing methods: SurCo and LANCER. We present the results for a randomly selected instance in Figure 6.

We have assigned a symbol of a corresponding S&P company to each point for clarity. On the Y-axis, we represent the expected return of a selected stock, which is multiplied by its fraction (*approximate* solution $\mathbf{x}$ in Eqn. (7)). Similarly, the X-axis displays the "risk−skewness score" calculated as follows: $\alpha \mathbf{x} \odot (\mathbf{G}\mathbf{x}) - \beta \mathbf{x} \odot (\mathbf{S}\mathbf{x} \otimes \mathbf{x})$, where $\odot$ denotes element-wise multiplication. This computation results in a vector of dimensions equal to the number of stocks, enabling us to interpret it as a score assigned to each stock. Furthermore, the combinatorial constraints outlined in Eqn. (7) enforce the selection of a maximum of $M$ stocks with fractions lower than or equal to $f_{\max}$.

As we try to maximize the return and minimize the risk−skewness score, we want all points to be in the upper-left corner. This is what LANCER achieves. It is interesting to see that LANCER chooses the less number of portfolios (6 vs 9) but assigns higher fraction to them improving overall objective.

# E  Stability of LANCER

Similar to many other approaches designed for solving problems with bi-level optimizations, including actor-critic algorithms, we currently lack theoretical guarantees regarding the convergence of LANCER with respect to either $\mathcal{M}_{\mathbf{w}}$ or $\mathbf{c}_{\boldsymbol{\theta}}$. This lack of guarantees can also impact the stability of the algorithm. However, this is primarily a matter that can be addressed through empirical investigation and depends on various factors, which is typical in the process of model selection. Nevertheless, we have observed that LANCER generally exhibits robustness in many cases. As an evidence, we performed additional experiments on nonlinear shortest path problem (section 5.1.2). In the two tables below we demonstrate that the final objective does not fluctuate significantly when we experiment with different combinations of hyperparameters and variations of neural net architecture. We will leave more ablation studies as the future work.

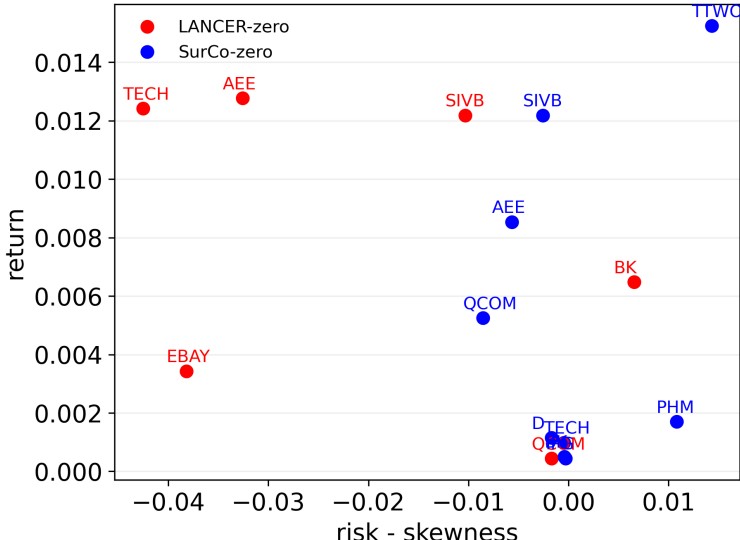

Figure 6: Visualization of the solution for a single instance of the combinatorial portfolio selection with third-order objective. LANCER demonstrates a tendency to select and allocate a significant proportion to stocks characterized by a combination of low 'risk − skewness' and high rewards (upper-left corner).

Table 4: Exploring various hyperparameters for LANCER for nonlinear shortest path problem from section 5.1.2.

| $\mathcal{M}_{\mathbf{w}}$ lrn_rate | $\mathcal{M}_{\mathbf{w}}$ max_itr | $\mathbf{c}_{\theta}$ lrn_rate | $\mathbf{c}_{\theta}$ max_itr | OBJECTIVE |
|---|---|---|---|---|
| 0.0005 | 10 | 0.001 | 10 | 0.4651 |
| 0.001 | 10 | 0.001 | 10 | 0.4649 |
| 0.01 | 10 | 0.001 | 10 | 0.4650 |
| 0.0005 | 5 | 0.001 | 10 | 0.4590 |
| 0.001 | 5 | 0.001 | 10 | 0.4597 |
| 0.01 | 5 | 0.001 | 10 | 0.4651 |
| 0.0005 | 20 | 0.001 | 10 | 0.4651 |
| 0.001 | 20 | 0.001 | 10 | 0.4612 |
| 0.01 | 20 | 0.001 | 10 | 0.4651 |
| 0.001 | 10 | 0.0005 | 10 | 0.4612 |
| 0.001 | 10 | 0.01 | 10 | 0.4650 |
| 0.001 | 10 | 0.001 | 20 | 0.4648 |

Table 5: Exploring various model architecture for LANCER for nonlinear shortest path problem from section 5.1.2.

| $\mathcal{M}_{\mathbf{w}}$ num of hidden layers | $\mathcal{M}_{\mathbf{w}}$ layer size | OBJECTIVE |
|---|---|---|
| 1 | 50 | 0.4644 |
| 1 | 100 | 0.4646 |
| 1 | 200 | 0.4641 |
| 2 | 50 | 0.4646 |
| 2 | 100 | 0.4651 |
| 2 | 200 | 0.4651 |
| 3 | 50 | 0.4644 |
| 3 | 100 | 0.4650 |
| 3 | 200 | 0.4642 |

