# OpenReview forum: "Landscape Surrogate: Learning Decision Losses for Mathematical Optimization Under Partial Information"
_NeurIPS.cc/2023/Conference — NeurIPS 2023 poster_

### Official Review · Reviewer_7fCr · 2023-07-03

**Soundness:** 2 fair
**Presentation:** 2 fair
**Contribution:** 3 good
**Rating:** 5
**Confidence:** 5

**Summary:**

This paper considers the class of problems known as Smart Predict and Optimize (or Decision-Focused Learning) where the learning task consists in learning the parameters of an optimization model given some of their features. The difficulty comes from trying to include the optimization model into the learning pipeline, since it is hard to differentiate over the optimization model. The paper proposes to learn two optimization model: one to learn the objective of the optimization model and one to learn the coefficient of the optimization model. These are trained using an iterative algorithm that tries them in sequence. The paper applies this approach to the traditional benchmarks used in SPO and show that the approach has benefits overall.

**Strengths:**

Algorithm 1 is the main contribution of the paper. It is an elegant way to solve SPO problems in general.

**Weaknesses:**

1. The formalization really gets in the way in this paper. I will try to list these issues here

1.a  The model M which approximates the solver should receive only one set of parameters. I do not understand why you would use c_\theta(y_i) and z_i in (3). It would be better to explain in detail the inputs of the model M

1.b In (4), the notation M_w(Y,Z;\theta^*) does not make any sense. M_w does not receive \theta^* in Algorithm 1. As a result, the bilevel model (4) does not make sense.

1.c why is the learning task expressed as min_{\theta,w} | M_w(c_theta(y)) - f(z) |?

2. Algorithm 1 is an ADMM approach to the optimization as shown above in fact.

3. I do not understand 4.1. The goal of SPO is to find the parameters of the optimization which is what c_\theta does? How is Model M useful on testing instances and model deployment? It does not compute any solution, just the objective.

4. In  4.2, you mention that you need Z to train M. This is contradictory with 4.1 where you are using it in unseen test instances. You do not have Z on unseen instances.


**Questions:**

See above.

**Limitations:**

The benefits on the portfolio optimization are computational only.

This area seems to always consider the same three problems.

---

> ### Author Rebuttal · Authors · 2023-08-07
>
> We sincerely thank the reviewer for providing valuable comments. While we partially agree with the summary provided by the reviewer, it is essential to note that LANCER is not only applicable to SPO problems but also to another crucial class of problems: solving mixed-integer nonlinear programming via learning surrogates. We unify learning surrogates for MINLP and SPO within the same learning framework, which is another important contribution that we would like to emphasize.
>
> According to the reviewer's concerns, the formalization of our approach (section 4) and Algorithm 1 require more clarification, which we address below. Moreover, we will incorporate most of these responses/clarifications into the revised version.
>
>
> 1. **Weaknesses**
>
> > 1.a. "The model M which approximates the solver should receive only one set of parameters. I do not understand why you would use c_\theta(y_i) and z_i in (3). It would be better to explain in detail the inputs of the model M"
>
> It is important to note that our main objective is to approximate composition function $f \circ \mathbf{g}$ (not just $f$ and NOT the solver $\mathbf{g}$ alone, but jointly), please check the main rebuttal for more detailed explanation. For this, evaluating $f \circ \mathbf{g}$ depends on both $c_\theta(y_i)$ and $z_i$. Therefore, to construct an accurate surrogate model $M_w$ that can effectively approximate $f \circ \mathbf{g}$, it is essential for $M_w$ to take into consideration the influence of both inputs. This consideration ensures that the surrogate captures the intricacies of the composition function and produces reliable predictions to aid the optimization process in both the smart P+O and learning surrogates for MINLP settings.
>
> > 1.b. "In (4), the notation M_w(Y,Z;\theta^*) does not make any sense. M_w does not receive \theta^* in Algorithm 1. As a result, the bilevel model (4) does not make sense."
>
> We apologize if it caused much confusion. The complete and more explicit form of Eqn. 4 should look like this:
>
> $\min_{w}{ \sum_i || M_w ( c_{\theta^*}(y_i), z_i ) - f( \mathbf{g}(c_{\theta^*}(y_i)), z_i ) || }$
> &nbsp;
>
> s.t. $\theta^* \in $ arg$min_\theta \sum_i M_w (c_ \theta(y_i), z_i) $
>
> As mentioned above in 1.a., the surrogate model $M_w$ takes $c_\theta(y_i)$ as one of its inputs. In this context, we are not concerned with making $M_w$ accurate for all possible $c_\theta$; rather, we are interested in finding the configuration of $M_w$ that yields the closest approximation to $f \circ \mathbf{g}$ near $\theta^*$. By adopting the bilevel optimization framework, we can focus on learning accurate $M_w$ for such $\theta$. In Algorithm 1, we achieve this by first evaluating $f \circ \mathbf{g}$ on the current "optimal" $c_{\theta^*}$, followed by retraining $M_w$ (i.e., $w$-step). Correspondingly, $w$-step ensures that $c_\theta$ optimizes the current surrogate loss $M_w$. We will make this more explicit in the paper and will add detailed description.
>
> > 1.c. "why is the learning task expressed as min_{\theta,w} | M_w(c_theta(y)) - f(z) |?"
>
> Hopefully, our response above (1.b.) also clarifies this matter. Otherwise, please do not hesitate to follow up during author-reviewer discussion phase.
>
> > 2. "Algorithm 1 is an ADMM approach to the optimization as shown above in fact."
>
> Although it has similar flavor, Algorithm 1 is not an ADMM because it does not follow the typical ADMM structure and optimization approach.
>
> ADMM is used to solve problems with separable objective functions and constraints. It is based on the idea of splitting the original problem into subproblems that can be solved in parallel, and then iteratively updating the variables using the method of multipliers. The general form of ADMM involves introducing auxiliary variables, Lagrange multipliers, and penalty terms to convert a constrained optimization problem into a series of subproblems that can be solved independently and in parallel. Clearly, the formulation in eq. (4) and Algorithm 1 do not exhibit this characteristic structure of ADMM.
>
> > 3. "I do not understand 4.1. The goal of SPO is to find the parameters of the optimization which is what c_\theta does? How is Model M useful on testing instances and model deployment? It does not compute any solution, just the objective."
>
> $M_w$ can be reused when Z = Y, i.e., there is no partial information in the testing instances, such as MINLP. In this case, $M_w$ can be used to evaluate the quality of target mapping $c_\theta(y)$, and improve $c$ directly (through $\theta$-step only) without calling the expensive composition $f \circ \mathbf{g}$. This is an optional step that one can perform once Algorithm 1 finishes. (see Algorithm 2 in the supplementary materials and refer to lines 143-149 in the main paper). This leads to a significant runtime reduction. We conducted ablation studies in section 5.4 to validate this.
>
> > 4. "In 4.2, you mention that you need Z to train M. This is contradictory with 4.1 where you are using it in unseen test instances. You do not have Z on unseen instances.
>
> If Z only contains partial information (e.g., P+O) during testing, we may still leverage M for "similar problems", providing computational advantages. For example, after using Algorithm 1 for one dataset of the portfolio selection (PS) problem, we can use the pre-trained M when training on another PS dataset either as a warm start or by executing the $\theta$-step. However, we did not validate this in our experiments, which may encounter potential issues such as distribution shifts and domain differences.

---

> > ### Comment · Reviewer_7fCr · 2023-08-11
> > **Rebuttal**
> >
> > I read the rebuttal. I believe the formalization is probably correct this time. I will check further. There are a lot of mistakes and unclear statements in this paper, as seen in the rebuttal. But I believe that the algorithm is interesting.

---

> > > ### Author Response · Authors · 2023-08-11
> > >
> > > Thank you for taking the time to review the rebuttal. Regarding the perceived mistakes and unclear statements, we'd like to clarify that these issues mainly stem from notational choices that inadvertently caused misunderstandings rather than fundamental issues. We are sorry for the imperfect presentation. Many of these concerns have been addressed in the rebuttal, and we will update our paper in the next revision. Let us know if you have more questions!

---

### Official Review · Reviewer_Nect · 2023-07-04

**Soundness:** 3 good
**Presentation:** 4 excellent
**Contribution:** 3 good
**Rating:** 7
**Confidence:** 5

**Summary:**

This paper proposes the a novel framework for learning the predicted value of an optimization task under limited information. Specifically, the goal is to learn $\mathcal{M}\_w(y, z) = f (g\_\theta (y), z)$ function where $y$ is the limited information, $z$ is the complete information, $f$ is the objective value of an optimization function to be maximized, and $g\_\theta$ is a way to generate a solution to the optimization problem based on the limited information $y$. In essence, there are 2 sets of interdependent parameters to learn -- $\theta$ and $w$, and the paper proposes an alternating update scheme for solving the joint problem. They then apply this framework to 2 classes of problems---Smart Predict-then-Optimize (SPO) and Mixed Integer Non-Linear Programming (MINLP)---and show that this method outperforms similar approaches in each class.

**Strengths:**

* **Great exposition**: The paper is well written and easy to follow.
* **Good experiments**: The paper uses domains from the literature and compares to relevant past work for each problem class. The experiments are well documented and seem reasonably expansive.
* **Interesting connection**: I'm more familiar with the SPO literature, so the connection of learned surrogates to MINLP is quite interesting!

**Weaknesses:**

The major weakness of this paper, imo, is that it doesn't document the **training considerations** for $\mathcal{M}_w$. The paper mentions in the conclusions that 'one potential drawback is the complexity of tuning M, requiring model selection and training', but there are no experiments about how hard/important tuning these parameters is. Specifically, it would be good to know:
1. Shah et al. [36] highlight the importance of *convex* surrogates for learning loss functions (because they have to be optimized over) and show specifically on a version of the Portfolio Optimization domain that neural networks performs poorly (Table 1 in their paper). However, in this paper $\mathcal{M}_w$ seems to be a neural network and performs well? Do you not observe this phenomenon of 'convexity being important for loss functions'?
1. There seems to be a recent follow up to [36] --> [A] which seems to do 1 iteration of your alternating update (with a 2-stage warm start) and seems to do well on the Portfolio Optimization domain, even better than the MDFL method (which seems to do much better than both LODLs and LANCER in this paper). Then:
    * How robust are your results to hyperparameter choices?
    * How important is it to perform repeated updates? (Is this what you mean by "epoch" in Figure 5? And if so, how much difference is there between 1 epoch with 10x the samples vs. 10 epochs with 1x the samples?)
    * Relatedly, how do those results change based on how good the initial guess of $\theta$ is? Do you need fewer updates if you warm-start $\theta$ from a 2-stage solution?
1. Why do you measure cost in terms of the number of calls to the BB optimizer in Figure 5? There seems to be a trade-off between (a) using more calls the BB optimizer but training a simple model (as in LODLs) vs. (b) using fewer calls to the BB optimizer but training a more complex model. Do the results look very different if you use wall-clock time?

_References:_

[A] Shah S, Perrault A, Wilder B, Tambe M. Leaving the Nest: Going Beyond Local Loss Functions for Predict-Then-Optimize. arXiv preprint arXiv:2305.16830. 2023 May 26.

**Questions:**

Apart from the questions in the weaknesses in the section above, I was wondering:
1. **Choice of domains**: I noticed that you have not used the domains from either SurCO or LODLs on which they report good performance. While this is not a weakness in itself, I can't help but wonder how well LANCER performs on those domains. Should LANCER always be the choice of learning method or is the answer more nuanced?
1. **Alternating updates**: Is it important to always do full updates in both spaces? Have you considered, perhaps, doing smaller updates? For example, in a recent paper, [B] seems to do some sort of meta-gradient update based on how $\theta$ would change in response to $w$.
1. **Re-using $\mathcal{M}_w$**: The fact that landscape surrogates can be 're-used' seems to suggest that $\mathcal{M}_w$ is learning something that is common across problem instances. Have you tried to analyze what it is that $\mathcal{M}_w$ is learning?

From the experiments, I'm convinced that there exist reasonable domains for which its possible to get this method to work. However, it's not clear at the moment how easy this process is... I'm willing to raise my score to a 7 if the authors provide more detailed answers to the questions in the weaknesses section

_References:_

[B] Sivasubramanian, Durga, et al. "Adaptive Mixing of Auxiliary Losses in Supervised Learning." Proceedings of the AAAI Conference on Artificial Intelligence. Vol. 37. No. 8. 2023.

**Limitations:**

I appreciate the inherent conflict of interest, but I think the paper could do a better job of documenting the limitations of their approach. For example, if the authors could answer (1) from the questions section above, that would help to better contextualize the advantages and limitations of this approach.

---

> ### Author Rebuttal · Authors · 2023-08-07
>
> 1. **Weaknesses**
>
> > 1.a."Do you not observe this phenomenon of 'convexity being important for loss functions'?"
>
> Please check the main rebuttal (points 2 and 3).
>
> > 1.b. Regarding follow up to LODLs.
>
> Thanks for the reference. We were unaware of this work as it appeared after the submission deadline to NeurIPS. As for the results on Portfolio Optimization, there are likely large differences in the data and evaluation. Specifically in the follow-up paper [A], the 2-stage method performs catastrophically poorly compared to LODLs and the proposed approach whereas in our experiments we found that 2-stage performed worse but not disastrously bad. It is thus unclear whether the relative model performance would apply to both settings.
>
> It is also important to note that while our results are slightly worse than MDFL, LANCER is clearly performing better in terms of runtime, as shown in Fig. 5 and acknowledged by Reviewer 7fCr.
>
> > 1.c. "How robust are your results to hyperparameter choices?"
>
> Please see our response to all reviewers above (point 1) for additional experiments we conducted to test the stability of our approach across various hyperparameters and neural network architectures.
>
> > 1.d. "How important is it to perform repeated updates? (Is this what you mean by "epoch" in Figure 5? And if so, how much difference is there between 1 epoch with 10x the samples vs. 10 epochs with 1x the samples?)"
>
> Yes, in Figure 5, we refer to an epoch as one step of alternating optimization. We find it quite important, and as shown in Figure 5, a larger $T$ (number of updates/iterations) typically leads to better performance. This is also related to our response to Question 2.b. below (making small number of inner updates but keeping larger $T$).
>
> As for the "1 epoch with 10x samples", this is prone to overfitting. Whereas, alternating optimization can help us to mitigate overfitting as the model sees the data multiple times with different parameter configurations. Additionally, if replay buffer (section 4.2.) is enabled, then we can naturally (re-)use the target mapping output to enlarge the number of samples. Intuitively, once the number of samples reach a certain threshold, performing more inner updates could be useful. This type of "adaptive" learning scheme we leave for future work. We empirically observed that performing repeated updates consistently improves the performance (around 1.5-2x times in accuracy).
>
> > 1.e. "Relatedly, how do those results change based on how good the initial guess of \theta is?"
>
> The quality of the initial guess (e.g. 2-stage) affects optimization results, with a good guess leading to faster convergence and better outcomes. For example, in nonlinear shortest path problem, random initialization needs 1.5x-2x more iterations to get the same performance.
>
> > 1.f. "Why do you measure cost in terms of the number of calls to the BB optimizer in Figure 5?"
>
> In all our experiments, the number of black-box (BB) accesses directly correlates with the runtime, with the w-step significantly dominating over the \theta-step (see Section 1 in the supplementary materials). This is especially true when solving high-dimensional combinatorial problems (e.g., Multidim knapsack, MINLP Portfolio Selection) where even highly optimized solvers (e.g., Gurobi) take time to solve for a single instance.
>
> On this note, LODLs as well take a long time during the sampling stage, where multiple calls to the optimizer occur. Additionally, a number of papers from the literature (e.g., [16]) report the number of BB calls as one of the main metrics.
>
> &nbsp;
>
> 2. **Questions**
>
> > 2.a. "Choice of domains: I noticed that you have not used the domains from either SurCO or LODLs on which they report good performance..."
>
> While we appreciate your observation, it's important to note that our focus was primarily on relatively high-dimensional problems, and we aimed to cover both domains of learning surrogates for MINLP and DFL. Running all experiments from both SurCO and LODLs papers proved challenging due to the extensive scope. However, it's worth mentioning that both papers reported good results on the benchmarks we tried. For examples, SurCO shows better or similar performance in some settings of nonlinear shortest path problem.
>
> > 2.b. "Alternating updates: Is it important to always do full updates in both spaces? Have you considered, perhaps, doing smaller updates? ..."
>
> In our approach, we perform a fixed (smaller) number of updates, i.e., we do not exactly solve the minimization in both spaces (lines 10 and 12 in Algorithm 1). For example, most experiment uses 10-20 updated per global iteration. Potential reason is that we don't want $\mathcal M_w$ model to overfit to the data much and increase $T$ instead. Although we discuss this in experimental setup, this will be made clear in the description of the algorithm to avoid any ambiguity.
>
>
> > 2.c. "Re-using $\mathcal M$. ... Have you tried to analyze what it is that is learning?""
>
> Indeed, the reuse of landscape surrogates suggests that the learning process captures common information shared across problem instances. While we haven't explicitly analyzed what the learning process captures, investigating the nature of the learned information could be an intriguing direction for further research.

---

> > ### Comment · Reviewer_Nect · 2023-08-12
> > **Response to Rebuttal**
> >
> > Thank you for the clarifications. I'm not convinced that the landscape surrogate is doing anything significantly different from other surrogate losses at a high-level (as long as $\mathbf{g^*}$ in LODLs and $\mathbf{g}$ in this paper are the same), but at the same time I think this paper addresses some key shortcomings of learning surrogate losses (sample efficiency and heuristic sampling). The experiments are quite thorough and seem to clearly show that the proposed improvements improve on past work. I think this paper is a very useful addition to the literature. I raise my score to 7.

---

### Official Review · Reviewer_kDSQ · 2023-07-06

**Soundness:** 2 fair
**Presentation:** 2 fair
**Contribution:** 2 fair
**Rating:** 5
**Confidence:** 4

**Summary:**

This paper presents a unified framework for "predict-then-optimize" and surrogate cost learning for Mixed Integer Nonlinear Programming (MINLP). These problems are cast as learning an optimizer g with f as the objective. Current solutions either suffer from scalability issues or the sparse gradient problem. To overcome these, the authors propose learning a smooth and tractable landscape surrogate to replace the compound function $f\circ g$. A neural network parameterizes the surrogate loss and it is learned through alternating optimization. This is done by alternately optimizing the target model c and landscape surrogate $\mathcal{M}$ in a manner similar to Generative Adversarial Networks (GANs). Experiments covering both linear and non-linear objectives demonstrate the efficacy of the proposed method under both "predict-then-optimize" and surrogate learning settings.

**Strengths:**

The paper proposes the first unified framework for "predict-then-optimize" and surrogate learning. The method is straightforward, and empirical results appear promising in terms of both optimization performance and runtime.

**Weaknesses:**

Directly utilizing a neural network to parameterize the landscape may not be a good idea. The complexity of a neural network typically surpasses the original objective f. A neural network can have extremely large number of local minimums and this approach could disrupt the convexity of the original optimization problem, leading to instability in the proposed method's learning procedure. In fact, in Shah et al., several parameterizations, including neural networks, were tested and found to often result in catastrophic outcomes. There's concern about this parameterization across various optimization objectives.

While SurCo minimizes the original objective f, LANCER learns an additional surrogate loss. However, in experiment 5.2.2, LANCER significantly outperforms SurCo. The authors should provide more analysis and clarification on why employing such a surrogate loss can surpass the original loss.

The paper lacks important implementation details. In 4.1, the authors mentioned the possibility of executing the theta-step during testing since c_{\theta} is available for unseen test data. It is unclear if this step was used during testing in the experiments. The supplementary material also does not provide details about the number of training iterations for this theta-step during testing. If this additional optimization step was used during testing, performance without this step should also be reported to pinpoint the source of the gains compared to SurCo. In fact, SurCo can also have this extra optimization step for the unseen test data.

Furthermore, the paper does not provide any implementation details for the replay buffer trick discussed in 4.2, such as the number of points needed to be stored, which could significantly impact the training time.

The unification of the "predict+optimization" problem and surrogate cost learning problem under optimization with partial information feels forced as half of the paper focuses on surrogate objective learning for MINLP where full information is available.

Section 4 could benefit from restructuring. The main introduction of the proposed method precedes 4.1 and 4.2 and lacks a subsection title, while 4.1 and 4.2 focus on implementation details.

Minor:

Line 285: MDFL was proposed for combinatorial decision-focused learning and it is not the first DFL paper. A more accurate citation should be Donti et al. (2017).

**Questions:**

Have different parameterizations of \mathcal{M} been tested and do they significantly affect performance?

In 4.1, the paper said we can execute the theta-step at testing time since c_{\theta} for unseen test data. Did you try this step in your experiment? I did not find the number of training iterations of this theta-step at testing time in the supplementarial material.

Was the replay buffer trick, as described in 4.2, used and how many points were needed to be stored? This could significantly impact training time but no implementation details are given in the experimental section or the supplementary material.

**Limitations:**

Please refer to the Weaknesses section for a detailed discussion of the paper's limitations.

---

> ### Author Rebuttal · Authors · 2023-08-07
>
> 1. **Weaknesses**
>
> > 1.a. On utilizing a neural network to parameterize the landscape and comparison with LODLs
>
> Please check our response to all reviewers above (points 2 and 3)
>
> &nbsp;
>
> > 1.b. Why LANCER outperforms SurCo?
>
> **LANCER allows nondifferentiable objective $f$**. SurCo relies on (approximately) differentiating through the solver, which can be challenging and slow in combinatorial problems due to sparsity, while LANCER circumvents this issue by constructing a surrogate landscape, M, that models the composition of nonlinear objective f *and* the solver g. M is differentiable and thus provides dense gradients and facilitates faster learning.
>
> **LANCER may call solver g fewer times**. SurCo requires that the solver $\mathbf{g}$ be called in each iteration, which can be time-consuming, while LANCER can leverage surrogate landscape $\mathcal M_w$ to bypass the solver $\mathbf{g}$, saving computational cost.
>
> The pros and cons of both approaches are thoroughly discussed in sections 3 and 4.
>
> > 1.c. Implementation details for section 4.1.
>
> We apologize for any confusion. The implementation of section 4.1 simply involves executing $\theta$-step for unseen data once Algorithm 1 terminates. This is especially useful when Z = Y, i.e., there is no partial information in the testing instances, such as MINLP (see LANCER-zero pseudocode in the suppl. mat.). In this case, $M_w$ can be used to evaluate the quality of target mapping $c_\theta(\mathbf{y})$, and improve $c$ directly (through $\theta$-step only) without calling the expensive composition $f \circ \mathbf{g}$. This is an optional step that one can perform once Algorithm 1 finishes.
>
> As for the experiments, we found out that this can lead to a significant runtime reduction. We conducted ablation studies in section 5.4 to validate this.
>
> > 1.d. Implementation details of the replay buffer and the effect on runtime.
>
> We primarily employed replay buffer for MINLP experiments, and it showcased noticeable improvements in performance. We set the number of points to be added to the buffer to N, so the total/max size of the replay buffer progressively reaches TN by the termination of the algorithm, where T is typically between 5-40. Of course, this could be controlled via setting the max buffer size if N is large. This is easy to implement as insert/retrieve operation on a buffer takes constant time. Additionally, we will make our implementation open source to allow full reproducibility.
>
> It is crucial to note that the use of the replay buffer has only a marginal effect on the runtime and is barely noticeable. The primary purpose of the replay buffer is to store and reuse past experiences to break temporal correlations and stabilize training. This process is relatively lightweight compared to other computationally intensive parts of the algorithm, such as interacting with the black-box solvers. The runtime increase only appears at line 10 of Algorithm 1, which involves training neural networks that scale linearly with the dataset size. Fortunately, this process is straightforward to parallelize, thanks to the capabilities offered by modern frameworks.
>
> &nbsp;
>
> > 1.e. "The unification of the P+O and surrogate cost learning feels forced..."
>
> We respectfully disagree with this statement. Our approach equally addresses both problems, each of which introduces uncertainty to the optimization problem. Neither of them can be straightforwardly solved in their general formulations. In the case of P+O, the problem descriptors are unknown and must be inferred from observed input $\mathbf{y}$. On the other hand, when learning surrogates for MINLP, the cost vector of the surrogate problem is unknown. We formulate both of these uncertainties as learning problems and propose LANCER as a unified algorithm to address them. The unification is crucial to derive LANCER and successfully apply it to both of these challenging problems, without changing substantially its framework.
>
> &nbsp;
>
> 2. **Questions**
>
> > 2.a. Have different parameterizations of $\mathcal M$ been tested and do they significantly affect performance?
>
> Please see point 3 in the main response.
>
> > 2.b. Experiments with regards section 4.1.
>
> The ablation study in section 5.4 is specifically dedicated to this purpose. Table 3 presents the results, clearly demonstrating the benefits of the "transferrability/reusability" of $\mathcal M_w$.
>
> > 2.c. Was the replay buffer trick, as described in 4.2, used?
>
> Please, see our response 1.d. in Weaknesses.

---

> > ### Author Response · Authors · 2023-08-18
> >
> > Dear Reviewer kDSQ,
> >
> > I hope our responses have resolved your concerns. We have diligently worked to address the points you raised and believe these revisions strengthen the overall quality of the paper. Your feedback has been invaluable in refining our work. If you find the revisions align well with the paper's objectives and address your initial concerns, we are hopeful that an adjustment in the score could reflect these improvements.
> >
> > Please feel free to ask if you have more questions or if there's anything else we can provide to support your evaluation. Thank you!

---

> > ### Comment · Reviewer_kDSQ · 2023-08-18
> > **Thanks for the rebuttal**
> >
> > Thanks for the detailed clarification. I have also read other reviewers. I will adjust my score accordingly.

---

### Official Review · Reviewer_1xim · 2023-07-07

**Soundness:** 3 good
**Presentation:** 3 good
**Contribution:** 3 good
**Rating:** 7
**Confidence:** 4

**Summary:**

`This paper is concerned with an amortized optimization scheme for challenging variants of canonical decision problems such as MINLPs and nonlinear portfolio selection. The authors propose a method with two components, a target mapping $c_\theta$ that maps partially observed problem descriptions $\mathbf y$ to full descriptions $\mathbf z$ that are passed to a traditional solver $g$ (e.g. an MILP solver), and a landscape estimator $M_w$ (i.e. a value function) that estimates the objective value attained by  $g(c_\theta(\mathbf y))$ w.r.t. some distribution of training instances. Together $g(c_\theta(\mathbf y))$ can be seen as a policy mapping problem descriptions to optimal decisions.
This is an empirical paper, with proof of concept results on several variants of canonical decision problems.

**Strengths:**

Amortized optimization is a common technique across machine learning. For example, LANCER bears striking similarity to actor-critic methods for RL, which amortizes the evaluation of an actor (i.e. policy) $\pi_\theta$ into a critic (i.e. value estimate) $Q_w$, and likewise the policy amortizes the optimization problem $a^*(s) = \mathrm{argmax}_a Q_w(s, a)$. The main difference is the objective value in RL (i.e. the policy returns) are not directly observed, and so $Q_w$ is trained by fitted Q-iteration (i.e. the Bellman backup operator) instead of a supervised loss as proposed in this paper.

While the technique proposed in this paper is not new, it is the first time I have seen it combined with traditional optimization solvers like MILPs, which are very underutilized by the ML community. Hence the concept of the paper is very intuitive, and has as much if not more potential for impact than RL papers, which are routinely published in NeurIPS proceedings but as yet have not appeared to have significant impact on industrial optimization problems.

**Weaknesses:**

Like many other solutions to bilevel optimization problems (including actor-critic algorithms), LANCER is not guaranteed to converge w.r.t. $\mathbf w$ or $\mathbf \theta$. Although it is not discussed in the paper, I suspect that LANCER is likely unstable if the hyperparameters are not chosen carefully. If this is not the case, please provide evidence in the rebuttal.

The connections to other amortized optimization methods in ML generally is not a weakness in my opinion, however I think the related work and discussion could be greatly improved by placing this work in context with other methods like actor-critic RL algorithms and amortized Bayesian inference algorithms to name a few.





**Questions:**

- How sensitive is LANCER to hyperparameters in terms of performance and stability?

- Do you think it may be necessary to regularize $c_\theta$ to better explore the space of possible target mappings (similar to entropy-regularized policies in RL e.g. soft actor-critic)?

**Limitations:**

- While LANCER appears more sample efficient in terms of black-box sample efficiency is does seem a bit slower than the baselines (although it does find better solutions). Hence in latency-sensitive applications it may not be the best choice

- Like Offline RL the generalization of LANCER to unseen problems will depend heavily on the support of that particular problem configuration in the training set. For out-of-distribution problems LANCER's performance is likely greatly diminished. Determining LANCER's sensitivity to distribution shift is an interesting research question.

---

> ### Author Rebuttal · Authors · 2023-08-07
>
> 1. **Weaknesses**
>
> > 1.a. Convergence guarantees and stability
>
> We acknowledge that, at present, we lack theoretical guarantees on convergence for LANCER. However, we observe empirically that the objective improves as the total number of alternating optimization iterations (T) increases, as shown in Figure 4.
>
> Also, please see our response to all reviewers above (point 1) for additional experiments we conducted to test the stability of our approach across various hyperparameters and neural network architectures.
>
> > 1.b. Connections to other amortized optimization methods
>
> We would like to extend our appreciation to the reviewer for highlighting the insightful parallels between the actor-critic framework and our LANCER methodology. Although we provide a review of several amortized methods in context of smart P+O and MINLP (e.g. [13,16,36]), we agree that enhancing the related work and discussion by placing our work in the context of actor-critic RL algorithms and amortized Bayesian inference algorithms, would be a valuable addition to the paper. We will certainly take your suggestion into consideration and make the necessary revisions to provide a more comprehensive understanding of the contributions and implications of our proposed approach in relation to these other methods.
>
> &nbsp;
>
> 2. **Questions**
>
> > 2.a. How sensitive is LANCER to hyperparameters in terms of performance and stability?
>
> See our response in the Weaknesses section (1.a.) above.
>
> &nbsp;
>
> > 2.b. "Do you think it may be necessary to regularize to better explore the space of possible target mappings (similar to entropy-regularized policies in RL e.g. soft actor-critic)?"
>
> In our experiments, we have chosen to utilize simple weight decay as a means of regularizing target mappings. Other regularization approaches could apply, e.g., dropout, sparsity, and/or use networks of fewer number of parameters. Note that entropy-based regularizations are more straightforward to apply when working with probability distributions, which might not be as straightforward in the context of our study, where problems in P+O & MINLP have a deterministic nature.
>
> &nbsp;
>
> 3. **Limitations**
>
> > 3.a. While LANCER appears more sample efficient in terms of black-box sample efficiency is does seem a bit slower than the baselines (although it does find better solutions). Hence in latency-sensitive applications it may not be the best choice
>
> In all our experiments, the number of black-box (BB) accesses directly correlates with the runtime, with the w-step significantly dominating over the \theta-step (see Section 1 in the supplementary materials). As indicated in Figure 5, LANCER consistently exhibits much faster runtime compared to the baseline methods. The only exception is shown in Figure 4, where we intentionally use more requests to the BB solver to explore a larger number of iterations for evaluation purposes.
>
> Moreover, the w-step in LANCER can be easily parallelized, offering the potential for significant improvements in runtime efficiency. We are confident that the parallelization capability can further enhance the overall performance of our approach.
>
> > 3.b out of distribution performance...
>
> On the portfolio optimization instances, the test data is quite out of distribution from the training data as the data are split temporally. In the financial setting, it is considered that the data distribution changes wildly over time. Our performance on this domain seems to suggest that the models are able to generalize to unseen data that are dissimilar to the training data. We leave more thorough studies for future work.

---

> > ### Comment · Reviewer_1xim · 2023-08-11
> > **Acknowledgement**
> >
> > Thanks for the thorough response! While there are areas for improvement that could likely increase the impact of the paper, I remain very supportive of acceptance.
> >
> > I'm very happy to see that your method does not appear too sensitive to hyperparameters. I really believe the practical impact of this paper could be significant.
> >
> > After reading some of the responses from the other reviewers, I think a challenge you are facing is the way new methods are tend to be held to a bit of a double standard compared to existing methods. Virtually all the criticisms you are facing also apply to every deep RL paper, and yet dozens of RL papers are happily accepted to major ML conferences every year. As you reflect on the relationship of your work to the field of ML as a whole you may find this tutorial on amortized optimization [1] provides a helpful perspective. You cite other papers by the same author several times in your paper so you may already be aware of it. In any case, I hope you find the exercise productive.
> >
> > My comment on the runtime of Lancer was primarily based on Figure 4, which seems to indicate LANCER has the longest runtime of the methods considered. How do you explain this apparent inconsistency?
> >
> >
> > [1] Amos, Brandon. "Tutorial on amortized optimization." Foundations and Trends® in Machine Learning 16.5 (2023): 592-732.
> > https://arxiv.org/abs/2202.00665

---

> > > ### Author Response · Authors · 2023-08-12
> > > **LANCER runtime clarification**
> > >
> > > Thank you for dedicating time to review the rebuttal and for contributing additional relevant research!
> > >
> > >
> > > > My comment on the runtime of Lancer was primarily based on Figure 4, which seems to indicate LANCER has the longest runtime of the methods considered. How do you explain this apparent inconsistency?
> > >
> > > The table below expands Figure 4, providing a more comprehensive LANCER vs. SurCo (top-performing methods) comparison via increased SurCo iterations. Despite LANCER's longer runtime, its objective consistently improves, unlike SurCo which plateaus. Thus, extending LANCER's runtime is justifiable. In this specific Figure 4 experiment, the observed runtime difference in LANCER is attributed to LANCER-zero relying on requisite sampling for surrogate loss training (Algorithm 2 in supplementary materials), leading to more black-box requests and increased runtime (which can be parallelized). This acknowledges LANCER's variable performance, including cases of extended runtime.
> > >
> > >
> > > ||LANCER|||SurCo||
> > > |----|--|--|------|--|--|
> > > | Iterations | Objective | Runtime | Iterations | Objective | Runtime |
> > > | 10  | -0.0429 | 81.12   | 10   | -0.0268 | 10.12 |
> > > | 20  | -0.0611 | 124.90  | 20   | -0.0373 | 18.25 |
> > > | 30  | -0.1000 | 220.05  | 100  | -0.0421 | 78.48 |
> > > | 40  | -0.1233 | 517.74  | 500  | -0.0423 | 253.30 |
> > > | 100 | -0.1609 | 2012.37 | 1000 | -0.0425 | 589.12 |

---

### Author Rebuttal · Authors · 2023-08-07

We sincerely thank the reviewers for providing valuable comments and taking the time to review our paper. Here we address questions raised by several reviewers.


> 1. [1xim,kDSQ,Nect] **Stability of LANCER to hyperparameters and NN architecture**

This is an empirical question and depends on various factors, as is typical in model selection. Nevertheless, we've observed that LANCER generally exhibits robustness in many cases. As an evidence, we performed additional experiments on the problem from section 5.1.2. In the two tables below we demonstrate that the final objective does not fluctuate significantly when we experiment with different combinations of hyperparameters and variations of NN architecture. We will leave more ablation studies as the future work. Also, see point 3 below on using quadratic losses.

|M_lrn_rate | M_max_itr	| c_lrn_rate | c_max_itr | OBJECTIVE|
|------------:|--------------:|------------:|-----------:|-------:|
|0.0005	   | 10	        | 0.001	     | 10	     | 0.4651|
|0.001	   | 10	        | 0.001	     | 10	     | 0.4649|
|0.01	   | 10	        | 0.001	     | 10	     | 0.4650|
|0.0005	   | 5	        | 0.001	     | 10	     | 0.4590|
|0.001	   | 5	        | 0.001	     | 10	     | 0.4597|
|0.01	   | 5	        | 0.001	     | 10	     | 0.4651|
|0.0005	   | 20	        | 0.001	     | 10	     | 0.4651|
|0.001	   | 20	        | 0.001	     | 10	     | 0.4612|
|0.01	   | 20	        | 0.001	     | 10	     | 0.4651|
|0.001	   | 10	        | 0.0005	 | 10	     | 0.4612|
|0.001	   | 10	        | 0.01	     | 10	     | 0.4650|
|0.001	   | 10	        | 0.001	     | 20	     | 0.4648|

|M_num_of_hidden_layers | M_layer_size | OBJECTIVE|
|:---------------------------:|------------:|-----------------:|
|1                      | 50           | 0.4644|
|1                      | 100          | 0.4646|
|1                      | 200          | 0.4641|
|2                      | 50           | 0.4646|
|2                      | 100          | 0.4651|
|2                      | 200          | 0.4651|
|3                      | 50           | 0.4644|
|3                      | 100          | 0.4650|
|3                      | 200          | 0.4642|

&nbsp;

> 2. [kDSQ,Nect] **Concerns on the form of landscape surrogate $\mathcal M$. Comparison between LANCER and other surrogate losses in P+O**

We acknowledge that there is indeed similarity between the Decision Loss (DL) in [36], Decision Quality (DQ) in [A] and our landscape surrogate $\mathcal M_w$. Using our notation, the DL/DQ consider the loss in the form of $\mathrm{DL}(\mathbf{\hat z}, \mathbf z) := f(\mathbf g^*(\mathbf{\hat z}(\mathbf y)), \mathbf z)$, where $\mathbf z$ is the ground truth specification of the problem instance, $ \mathbf{\hat{ z }} $ its estimate, $\mathbf y$ is the features, and $\mathbf g^*$ is the groundtruth solver, and propose to use a surrogate loss $L_\phi$ to fit $\mathrm{DL}(\mathbf{\hat z}, \mathbf z)$.

In contrast, our landscape surrogate $\mathcal M_w(\mathbf c, \mathbf z)$ is used to fit the loss in the form of $f(\mathbf g(\mathbf c(\mathbf y)), \mathbf z)$. Note that LANCER focuses on the quality of the final objective directly, rather than aim at a good estimate of $\mathbf z$ first. Therefore, there are a few key differences:

+ we never aim to predict a good estimate of $\mathbf z$ so our loss cannot be written as the form of $DL(\mathbf{\hat z, z})$ and thus do not have the same structure as in DL/DQ loss. In fact, from the function arguments, it is clear that $\mathcal M_w(\mathbf c,\mathbf z)$ is not a symmetric function. This can be advantageous, since what we want is the solution to $f$ and getting a good estimate of $\mathbf z$ is extra work.

+ we use a target mapping $\mathbf c(\mathbf y)$ to predict the surrogate cost for a cheap surrogate solver $\mathbf g$ to give a solution. Therefore, $\mathbf g$ may be very different from the original ground truth solver, which can be very slow or impossible to run (as in the case with MINLP), and our landscape surrogate $\mathcal M_w(\mathbf {c,z})$ needs to cope with that.

+ LODLs are trained per-instance on points sampled around the true labels which may not be representative of the region that the target model is passing through during training. Furthermore, in such a tight region it makes sense to use a simple convex model rather than a more expressive neural network. However, when trying to approximate the broad region that our target model traverses during training *across* a diverse training set (Algorithm 1, line 5), it is likely that a more expressive model is needed.

Therefore, in our case fitting it with a NN is reasonable and have more benefits:

- Expressiveness: NNs are capable of approximating complex functions, making them well-suited for approximating the Value- or Q-functions in actor-critic RL algorithms, where they are widely used. Similarly, in our setting, $f \circ \mathbf{g}$ can exhibit complex behavior, especially in high-dimensional problems. For instance, we observe such behavior in the P+O Multidim Knapsack and MINLP portfolio selection problems.

- Scalability: we mainly focused on relatively high-dimensional optimization problems, and NNs can efficiently handle such large-scale tasks, enabling effective learning in these challenging environments.

[A] Shah et al. Leaving the Nest: Going Beyond Local Loss Functions for Predict-Then-Optimize. arXiv:2305.16830.

> 3. **Other alternative landscape losses we tested**

Finally, we would like to emphasize that our method is not specifically tailored towards neural nets and is more generic. Any differentiable function $\mathcal M$ could be used. We conducted experiments with convex functions in several instances. For instance, in the nonlinear shortest path problem (section 5.1.2), the quadratic model yields an objective value of 0.412 (i.e., worse than a greedy approach). In contrast, the results obtained using the NNs are close to the optimum, with an obj value of 0.464.

---

### Decision · Program_Chairs · 2023-09-21

**Decision:**

Accept (poster)

**Comment:**

This paper develops a novel approach for learning the predicted value of an optimization task under limited information. In this problem, there are two set of inter-dependent parameters to learn and the approach considers alternating update to solve this joint learning problem. The approach is applied to two problem classes: smart predict-then-optimize and mixed integer non-linear programming, and demonstrated good results. The paper also addresses some key shortcomings of learning surrogate losses (sample efficiency and heuristic sampling).

Overall, reviewers' were positive about the paper and author rebuttal addressed their questions and concerns.

I recommend accepting the paper and strongly encourage the authors' to incorporate reviewers' comments and rebuttal discussion into the final paper.